# SHB1 and CCA1 interaction desensitizes light responses and enhances thermomorphogenesis

Qingbin Sun [1], Shulei Wang [1], Gang Xu[1], Xiaojun Kang[2], Min Zhang [1] & Min Ni [2]

Light and temperature are two important environmental signals to plants. After dawn, photo-activated phytochromes translocate into the nucleus and interact with a family of negative basic helix-loop-helix PIF regulators. Subsequent phosphorylation and degradation of PIFs triggers a series of photomorphogenic responses. However, excess light can damage the photosynthetic apparatus and leads to photoinhibition. Plants acclimate to a balanced state of photomorphogenesis to avoid photodamage. Here, we show that upregulation of *PIF4* expression by SHB1 and CCA1 under red light represents a desensitization step. After dawn, the highly expressed circadian clock protein CCA1 brings circadian signals to the regulatory region of the *PIF4* signaling hub. Recruitment of SHB1 by CCA1 modulates red light-specific induction of *PIF4* expression thus integrating circadian and light signals. As noon approaches and light intensity and ambient temperature tend to increase, the SHB1–CCA1 interaction sustains *PIF4* expression to trigger thermomorphogenic responses to changing light and temperature conditions.

[1] National Key Laboratory of Crop biology, College of life Sciences, Shandong Agricultural University, Taian 271018, China. [2] Department of Plant and Microbial Biology, University of Minnesota at Twin Cities, Saint Paul, Minnesota 55108, USA. Correspondence and requests for materials should be addressed to M.N. (email: nixxx008@umn.edu)

Light signal is an important abiotic environmental factor. The regulation of plant growth and development by light signals involves three major classes of photoreceptors, the red (R) and far-red (FR) light-absorbing phytochromes and the UV-A/blue light-absorbing cryptochromes and phototropins[1]. *Arabidopsis thaliana* has five phytochrome genes, *PHYA* to *PHYE*[2–4]. Phytochromes (phy) are photo-reversible between the red absorbing form (Pr) and the far-red absorbing form (Pfr). Following conversion to the biologically active Pfr form, phytochromes translocate into the nucleus and interact with an important subfamily of bHLH transcription factors, the phytochrome interacting factors (PIFs). Their interaction leads to their phosphorylation, ubiquitination, and degradation via the 26 S proteasome and alters gene expression rapidly[5–8]. PIFs play a variety of roles in regulating plant light responses, such as seed germination, seedling skotomorphogenesis, de-etiolation, shade avoidance, and flowering[9–11].

Photoreceptors and the circadian clock sense and integrate diurnal and seasonal changes in environmental signals and modulate plant growth and development. The circadian clock regulates adaptation of plants to the alternation of day and night. The initial model of the plant circadian clock is a feedback loop, including the central oscillator components circadian clock associated 1 (CCA1, an MYB transcription factor), late elongated hypocotyl (LHY, a homolog of CCA1), and timing of CAB2 expression 1 (TOC1) and its related pseudoresponse regulators (PRRs). CCA1 and LHY are the first two plant clock genes identified, and bind directly to the *TOC1* promoter and repress *TOC1* expression[12–15]. TOC1 is a DNA-binding transcriptional factor and functions as a general transcriptional repressor of clock genes including CCA1 and LHY[16–18]. Recently, a number of new components have been integrated into clock models. These include the reveille (RVE) family of MYB transcription factors, which act in a feedback loop as transcriptional activators[19,20]. In addition, light-regulated WD 1/2 (LWD1/2) and night light-inducible and clock-regulated 1/2 (LNK1/2) are also transcriptional activators involved in circadian clock function[21,22].

Light and temperature are the most dynamic parameters in plant growth and development. PIFs act as pivotal components in a cellular signaling hub integrating biotic and abiotic pathways to regulate plant growth[10]. For example, PIF4 mediates plant adaptation to elevated ambient temperature or thermomorphogenesis[23,24]. Developmental and morphological changes are induced by high ambient temperature below the heat stress range[25]. In *Arabidopsis*, the changes include increased elongation of hypocotyls and petioles, hyponastic growth, and development of thinner leaves[26]. In addition, *pif4* exhibits an early flowering phenotype compared with wild type under high ambient temperature[23]. PIF4 coordinates this response by activating hormonal modules that subsequently regulate growth. PIF4 interacts with brassinazole-resistant 1 (BZR1), a transcription factor induced by brassinosteroid, and activates the expression of a number of downstream genes that drive shoot organ elongation[27,28]. PIF4 also directly regulates auxin levels by activating several auxin biosynthesis genes such as *yucca 8* (*YUC8*) to promote hypocotyl elongation and hyponastic leaf growth[29,30]. TOC1, the evening-expressed circadian clock protein, directly interacts with PIF4 and prevents its activation of downstream target genes, thereby suppressing thermomorphogenesis specifically at the end of day and evening[24].

Although light signals are fundamental to the growth and development of plants, excess light energy damages the photosynthetic apparatus and frequently enforces an inhibitory effect on photosynthesis known as photoinhibition[31]. Light inevitably generates excess reactive oxygen species (ROS) can be generated during photosynthesis under strong light, which can lead to photoinhibition and oxidative damage to the photosynthetic apparatus[32–35]. Photosynthetic organisms are able to acclimate to different environmental conditions to alleviate the detrimental effects of excess light on growth and viability[36]. For example, under excess light conditions, plants may acquire a balanced state of photomorphogenesis to avoid the absorption of excess light energy and to reduce photodamage through a feedback mechanism. Although the molecular mechanisms stimulating response to light are well established, those required for desensitization of perceived light signals remain less understood.

*SHB1* was initially isolated from the gain-of-function mutant *short hypocotyl under blue 1 Dominant* (*shb1-D*) based on its long hypocotyl phenotype under red, far-red, and blue light[37]. In *shb1-D*, a T-DNA is inserted 129 base pairs upstream of the *SHB1* start codon and causes *SHB1* overexpression. SHB1 contains an N-terminal SPX domain and a C-terminal EXS domain homologous to yeast suppressor of yeast GPA1 (SYG1) family proteins[37]. Its N terminus retains the function of full-length SHB1, and over-accumulation of the SHB1 C terminus causes a dominant-negative phenotype[38]. *PIF4* expression is increased in *shb1-D* compared with wild type and decreased in *shb1* partial loss-of-function mutant specifically under red light[37]. The molecular mechanism by which SHB1 regulates *PIF4* expression specifically under red light and its biological implication are still unknown. In this study, SHB1 hijacks the highly expressed central oscillator component CCA1 and LHY in the morning and is targeted to the *PIF4* promoter. During the day when light intensity and temperature increase, the SHB1–CCA1/LHY interaction sustains *PIF4* expression in response to both red light and higher ambient temperature. This mechanism serves two important purposes: upregulate *PIF4* expression to desensitize light responses for optimal photomorphogenesis, and enhance plant thermomorphogenesis for better survival under elevated ambient temperature.

## Results

**SHB1, CCA1, and LHY regulate *PIF4* expression under red light**. In an early study, SHB1 specifically upregulated *PIF4* expression under red light as assessed by real-time quantitative PCR and RNA gel blot hybridization analysis[37]. In general, *PIF4* expression was induced by red light, downregulated in *shb1* and upregulated in *shb1-D* compared with wild type under red light (Fig. 1a). Although *PIF1*, *PIF3*, *PIF5*, and *PIF7* expression was induced by red light, only the red light-induced expression of *PIF7* was partially compromised in *shb1* but not in *shb1-D* (Supplementary Fig. 1a). The rhythmic expression of *PIF4* is controlled by the circadian clock[39–41]. We also examined the rhythmic expression of *PIF4* in Col, *shb1*, Ws, and *shb1-D* seedlings under continuous red light after growth under 12-hr dark and 12-hr 2 µmol m$^{-2}$ s$^{-1}$ red light for 7 days (Fig. 1b, c). Samples were obtained at ZT0 and every 3 h thereafter for 36 h. Both *shb1* and *shb1-D* mutations affected the magnitude but not the rhythmic pattern of *PIF4* expression.

When *PIF4* was driven by its native promoter, PIF4 protein accumulation correlated with *PIF4* transcription, and PIF4 protein accumulated during the light period from ZT0 to ZT8 but not in the dark period from ZT12 to ZT20 under short days (8 h white light/16 h dark)[42]. Under a 12-hr red light and 12-hr dark photoperiod, PIF4 protein was detectable after ZT3 and reached its peak accumulation at ZT9, correlating well with *PIF4* mRNA level (Fig. 1b–e). Before ZT3, the lower level of PIF4 accumulation was probably due to the low level of *PIF4* mRNA and/or PIF4 degradation after the initial illumination with red

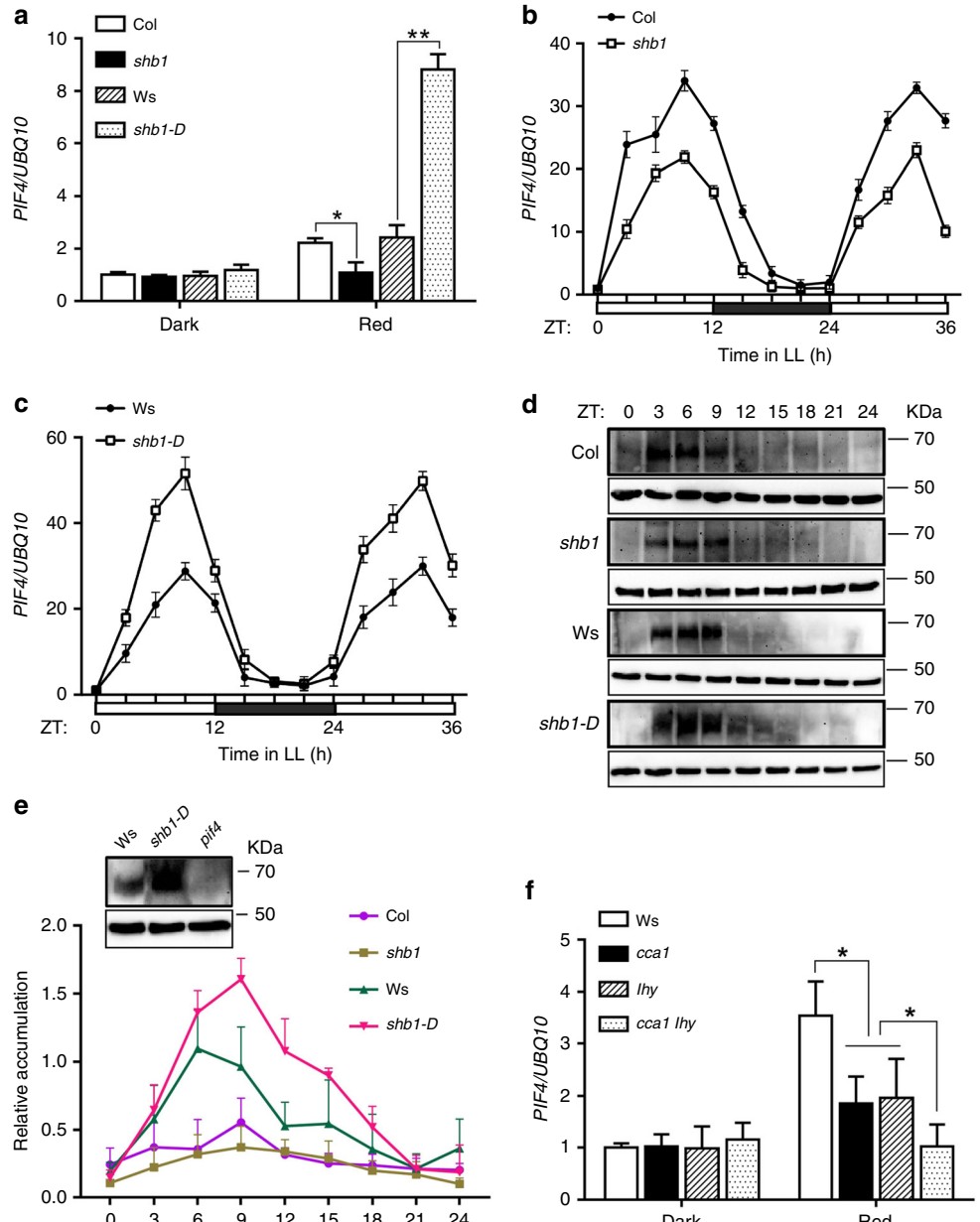

**Fig. 1** SHB1 and CCA1/LHY regulate *PIF4* expression under red light. **a** *PIF4* expression in 4-day-old Col, *shb1*, Ws, and *shb1-D* seedlings in the dark and under 15 µmol m$^{-2}$ s$^{-1}$ red light for 3 hr. * and ** indicate significance levels $p < 0.05$ and $p < 0.01$, respectively. ns indicates not significant by Student's two-tailed heteroscedastic *t* tests in this figure and subsequent figures. *PIF4* expression in Col and *shb1* **b** or Ws and *shb1-D* **c** under continuous red light after entrained under 12-hr light and 12-hr dark for 7 days from two biological replicates. **d** PIF4 protein accumulation in Col, *shb1*, Ws, and *shb1-D* under 12-hr red light and 12-hr dark photoperiod, shown as representative images from three biological replicates. **e** PIF4 protein quantification normalized to actin with standard error bars. The control blot shows PIF4 accumulation in Ws, *shb1-D* and *pif4* at ZT6. **f** *PIF4* expression in Ws, *cca1, lhy,* and *cca1 lhy* in the dark and under 15 µmol m$^{-2}$ s$^{-1}$ red light. Expression of *PIF4* in each sample was normalized to that of *UBQ10*, and data are presented as the means ± SE. Source data are provided as a Source Data file

light. PIF4 accumulation was strongly enhanced in *shb1-D* from ZT6 to ZT12 but moderately reduced in *shb1* from ZT6 to ZT12 and beyond (Fig. 1e).

SHB1 is unable to target to the *PIF4* locus given that it contains no recognizable DNA-binding domain. Other transcription factors likely recruit SHB1 to the *PIF4* promoter. We identified several potential MYB-binding elements ATATC(T/A) in the *PIF4* promoter (http://meme-suite.org/). In *Arabidopsis*, MYB family transcription factors have multiple functions[43]. The two central circadian clock components, CCA1 and LHY, are potential candidates. *PIF4* expression was reduced in the *cca1*

or the *lhy* single mutant, and was further reduced in the *cca1 lhy* double mutant (Fig. 1f). *PIF4* expression was not altered in *cca1* or *lhy* single mutants or in the *cca1 lhy* double mutant in the dark. The rhythmic expression of *PIF4* is controlled by the circadian clock[39–41]. In the *cca1 lhy* double mutant, the rhythmic expression of *PIF4* was completely compromised (Supplementary Fig. 1b). In addition, *SHB1* expression was not rhythmically expressed compared with *CCA1* and *LHY* (Supplementary Fig. 1c). Therefore, CCA1 and LHY are required for the rhythmic expression of *PIF4*, and SHB1 enhances but does not alter the pattern of *PIF4* rhythmic expression.

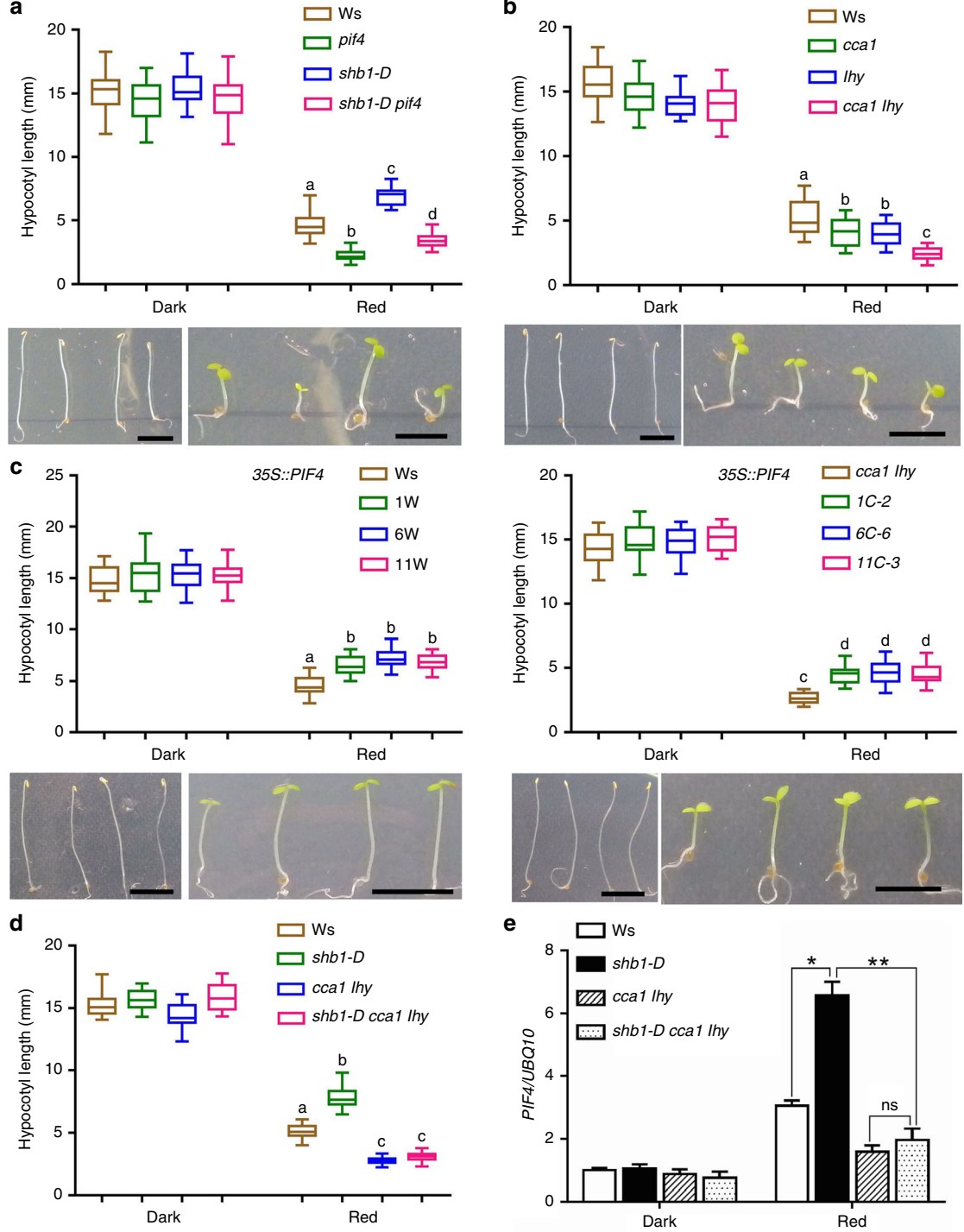

**Fig. 2** Genetic interaction of *PIF4* with *SHB1* or *CCA1/LHY*. Hypocotyl lengths of Ws, *pif4*, *shb1-D*, and *shb1-D pif4* **a**; Ws, *cca1*, *lhy*, and *cca1 lhy* **b**; three *35 S:: PIF4:GFP* (*pMDC83*) transgenic lines in Ws or *cca1 lhy* **c**; and Ws, *shb1-D*, *cca1 lhy*, and *shb1-D cca1 lhy* **d** in the dark or under 10 µmol m$^{-2}$ s$^{-1}$ red light for 4 days. Bar = 5 mm. Box plots display medians as horizontal lines, interquartile ranges as boxes and whiskers extending 1.5 times the interquartile range in this and subsequent figures. Significance levels by Student's two-tailed heteroscedastic *t* tests in a: *p* < 0.001 between **a** and **b**, **a** and **c**, **a** and **d**, **b** and **d**, or **c** and **d**; in **b**: *p* < 0.01 between **a** and **b**, *p* < 0.001 between **a** and **c** or **b** and **c**; in **c**: *p* < 0.001 between **a** and **b** or **c** and **d**; in **d**: *p* < 0.001 between **a** and **b**, **a** and **c** or **b** and **c**. **e** *PIF4* expression in Ws, *shb1-D*, *cca1 lhy*, and *shb1-D cca1 lhy*. *PIF4* expression in each sample was normalized to that of *UBQ10*, and data are presented as the means ± SE. Source data are provided as a Source Data file

## Genetic interaction between *PIF4* and *SHB1*, *CCA1*, and *LHY*.

We investigated the genetic interaction between *SHB1* and *PIF4* by measuring the hypocotyl elongation of Ws, *pif4*, *shb1-D* and the *shb1-D pif4* double mutant under red light. *shb1-D* exhibited a long hypocotyl, and *pif4* exhibited a short hypocotyl compared

with wild type. The *shb1-D pif4* double mutant exhibited a slightly longer hypocotyl compared with *pif4*, but it was considerably much shorter than that of *shb1-D* (Fig. 2a). The *shb1-D* mutation still increased hypocotyl length in the *pif4* mutant background or SHB1 also promoted hypocotyl elongation independent of PIF4,

SHB1 may target to other genes in addition to *PIF4* to influence hypocotyl elongation.

Although *CCA1* and *LHY* are redundant genes, the *cca1* or *lhy* single mutant exhibited a short hypocotyl similar to that of *pif4*, and the *cca1 lhy* double mutant exhibited a considerably shorter hypocotyl compared with each single *cca1* or *lhy* mutant (Fig. 2b). To study their epistasis, we overexpressed *PIF4:GFP* driven by the CaMV 35 S promoter in Ws and crossed three independent transgenes to the *cca1 lhy* background (Supplementary Fig. 2a). The hypocotyl of the *35 S::PIF4:GFP* transgenic lines was considerably elongated in Ws (Fig. 2c). Overexpression of *PIF4::GFP* in *cca1 lhy* generated a shorter hypocotyl compared with that in Ws, but a considerably longer hypocotyl compared with the *cca1 lhy* double mutant. Combining the results demonstrating that the red light-induced *PIF4* expression relies on CCA1 and LHY (Fig. 1f and Supplementary Fig. 1b), we conclude that PIF4 acts downstream of CCA1 and LHY.

The *shb1* mutation caused a minimal hypocotyl phenotype but affected *PIF4* expression under red light[37]. The *shb1* mutant allele also showed a partial loss-of-function phenotype in endosperm proliferation and cellularization[44,45]. Triple *shb1 cca1 lhy* mutant was constructed, and the hypocotyl length and *PIF4* expression were barely affected when *shb1* was introduced to *cca1 lhy* (Supplementary Fig. 2b, c). In *shb1*, a T-DNA is inserted at the 8th intron of the *SHB1* gene (SALK_128406), and a truncated message was still produced (Supplementary Fig. 2d, e). Given the extremely low level of *SHB1* expression and the low sensitivity of Taq polymerase used for semi-quantitative PCR analysis, we previously failed to detect the truncated message[37]. The full-length SHB1 protein contains 745 amino acids, and *shb1* lacks the C-terminal 223 amino acids but retains the N-terminal 522 amino acids. Given that the N terminus of SHB1 is important for its function[38], the truncated SHB1 in *shb1* is partially functional. In contrast, a *shb1-D cca1 lhy* triple mutant showed a hypocotyl phenotype and *PIF4* expression similar to that of the *cca1 lhy* double mutant (Fig. 2d, e and Supplementary Fig. 2f). We examined the expression of *SHB1* in *shb1-D* and *shb1-D cca1 lhy* triple mutant (Supplementary Fig. 2g). The levels of *SHB1* transcripts are comparable in *shb1-D* and *shb1-D cca1 lhy*. Thus, the promotion of hypocotyl and *PIF4* expression by SHB1 under red light critically relies on CCA1 and LHY.

**Regulation of PIF4 expression enhances thermomorphogenesis.** Given that PIF4 is required for thermomorphogenesis, the induction of *PIF4* expression by SHB1 and CCA1 may affect plant thermomorphogenesis. We examined hypocotyl growth responses of Ws, *shb1*, *pif4*, *shb1-D,* and the *shb1-D pif4* double mutant to warm temperatures. Col or Ws showed a clear thermos-response and this response was slightly reduced in *shb1* but almost disappeared in *pif4* (Fig. 3a). *shb1-D* enhanced the thermos-response with an elongated hypocotyl, and this enhancement was not completely blocked in *shb1-D pif4* double mutant. Therefore, the thermo-induced hypocotyl elongation through PIF4 under higher temperature may not be entirely dependent on SHB1.

Furthermore, warm temperature-induced hypocotyl growth of either *cca1* or *lhy* was similar to that of Ws, and *cca1 lhy* double mutations indeed reduced plant responses to warm temperatures (Fig. 3b). Overexpression of PIF4 driven by the *35 S* promoter enhanced the thermo-response in either Ws wild type or *cca1 lhy* backgrounds (Fig. 3c and Supplementary Fig. 3). PIF4-overexpression plants grown at 20 ºC exhibited a longer hypocotyl and increasing the temperature to 29 ºC caused a further elongation of the hypocotyl. However, the ratio of hypocotyl length at 20 ºC versus 29 ºC in PIF4-overexpression

plants was comparable in either a wild type or *cca1 lhy* double mutant background. Warm temperature-induced *PIF4* expression was reduced in *shb1* and *cca1 lhy*, but significantly enhanced in *shb1-D* compared with that in Col or Ws wild type (Fig. 3f).

Hypocotyls of the *cca1 lhy* double mutant were elongated in response to high temperature, and *PIF4* expression was still upregulated under high temperatures in *cca1 lhy*. The degree of upregulation was comparable between the wild type and the mutants. Therefore, the induction of the PIF4-mediated thermos-response under high temperature may not completely rely on CCA1 and LHY. In general, SHB1 and CCA1/LHY may not be required for thermos-activation of PIF4, but the overall increase in *PIF4* transcription increases the magnitude of the thermo-response. Both hypocotyl and *PIF4* expression at 29 ºC in *shb1-D cca1 lhy* triple mutant were higher than that of *cca1 lhy* double mutant, but lower than that of *shb1-D* (Fig. 3d–f). In addition to CCA1/LHY, other transcription factors may participate in SHB1-meditated thermos-responses.

**SHB1, CCA1, and LHY associate with the *PIF4* promoter.** We assessed whether SHB1 directly regulates *PIF4* expression although SHB1 does not contain a recognizable DNA-binding motif. We performed ChIP experiments with a SHB1:GFP transgene driven by the 35 S promoter and anti-GFP antibody followed by quantitative PCR (qPCR) analysis. We used four pairs of primers that detect different regions in the *PIF4* promoter (Fig. 4a). The samples were collected at ZT3 (8 am as ZT0) for dark-grown plants without or with 3 h of red light treatment. SHB1 was associated with 4–3 fragment in the *PIF4* promoter (Fig. 4b). Slight enrichment was observed for the 4–4 genomic fragment but the results were not statistically significant. The SHB1:GFP construct caused a longer hypocotyl phenotype and elevated *PIF4* expression (Supplementary Fig. 4a, b).

Two MYB-binding sites, M1 and M2, were identified in these two fragments (Fig. 4a). MYB transcription factors CCA1 or LHY may mediate the association of between SHB1 and the *PIF4* promoter. We performed ChIP experiments with CCA1:FLAG and LHY:MYC transgenes driven by the 35 S promoter and anti-FLAG or anti-MYC antibodies, respectively. CCA1 and LHY were also associated with the *PIF4* promoter, and the 4–3 fragment was enriched under red light (Fig. 4c and Supplementary Fig. 4c). We further examined whether SHB1 remained associated with the *PIF4* promoter if CCA1 and LHY are mutated. SHB1 was unable to associate with the 4–3 genomic fragment in the *PIF4* promoter in *cca1 lhy* (Fig. 4c). We examined the accumulation of SHB1:GFP in Ws and *cca1 lhy*, and their levels were comparable in either Ws or *cca1 lhy* in the dark and under red light at the time points sampled for ChIP analysis (Supplementary Fig. 4d). SHB1:GFP overexpression caused a longer hypocotyl phenotype and elevated *PIF4* expression under red light in Ws but not in *cca1 lhy* (Supplementary Fig. 4e, f).

CCA1:FLAG and LHY:MYC did not associated with the *PIF4* promoter in the dark (Fig. 4c and Supplementary Fig. 4c). CCA1:FLAG or LHY:MYC levels were comparable in the dark and under red light for 3 h when we sampled for ChIP analysis (Supplementary Fig. 4d). LUX, a component of the evening complex (EC), recognizes a consensus element GATWCG and two different types of degenerate elements, GATWYG or GATWCK[46]. The 4–4 fragment has one consensus element GATTCG that is 210 bp from the CCA1/LHY-binding element in the *PIF4* promoter. The 4–3 fragment contains one degenerate element, GATTCC, which is 90 bp away from the CCA1/LHY-binding element. In addition, another degenerate element, GATTTG, is 1 bp away from the forward primer of the 4–3 fragment. In the EC complex, LUX interacts with ELF3 and ELF3

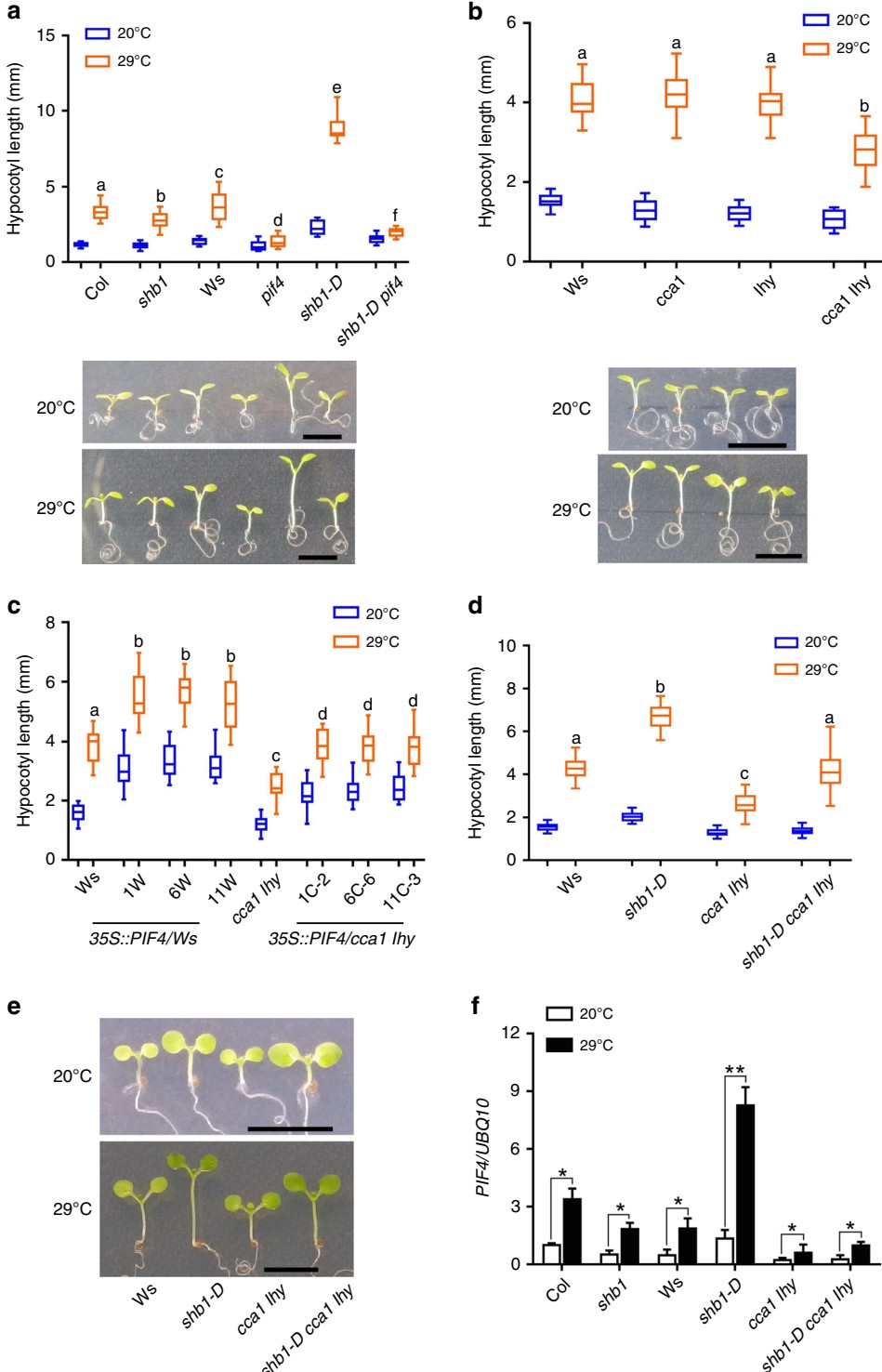

**Fig. 3** CCA1/LHY and SHB1 enhance *PIF4* expression and thermomorphogenesis. Hypocotyl lengths of Col, *shb1*, Ws, *pif4*, *shb1-D*, and *shb1-D pif4* **a**; Ws, *cca1*, *lhy*, and *cca1 lhy* **b**; three *35S::PIF4:GFP* lines in Ws or *cca1 lhy* **c**; and Ws, *shb1-D*, *cca1 lhy*, and *shb1-D cca1 lhy* **d**, **e** under 30 μmol m$^{-2}$ s$^{-1}$ white light at 20 °C for 7 days or 20 °C for 4 days followed by 29 °C for 3 days. Bar = 5 mm. Significance levels by Student's two-tailed heteroscedastic *t* tests in a: $p < 0.001$ between **a** and **b**, **c** and **d**, **c** and **e**, **c** and **f**, **d** and **e**, or **e** and **f** and $p < 0.01$ between **d** and **f**; in b: $p < 0.001$ between **a** and **b**; in c: $p < 0.001$ between **a** and **b** or **c** and **d**; in d: $p < 0.001$ between **a** and **b**, **a** and **c** or **b** and **c**. **f** *PIF4* expression in Col, *shb1*, Ws, *shb1-D*, *cca1 lhy*, and *shb1-D cca1 lhy*. Seedlings were grown at 20 °C for 5 days and then incubated at 20 °C or 29 °C for 4 h. *PIF4* expression in each sample was normalized to that of *UBQ10*, and data are presented as the means ± SE. Source data are provided as a Source Data file

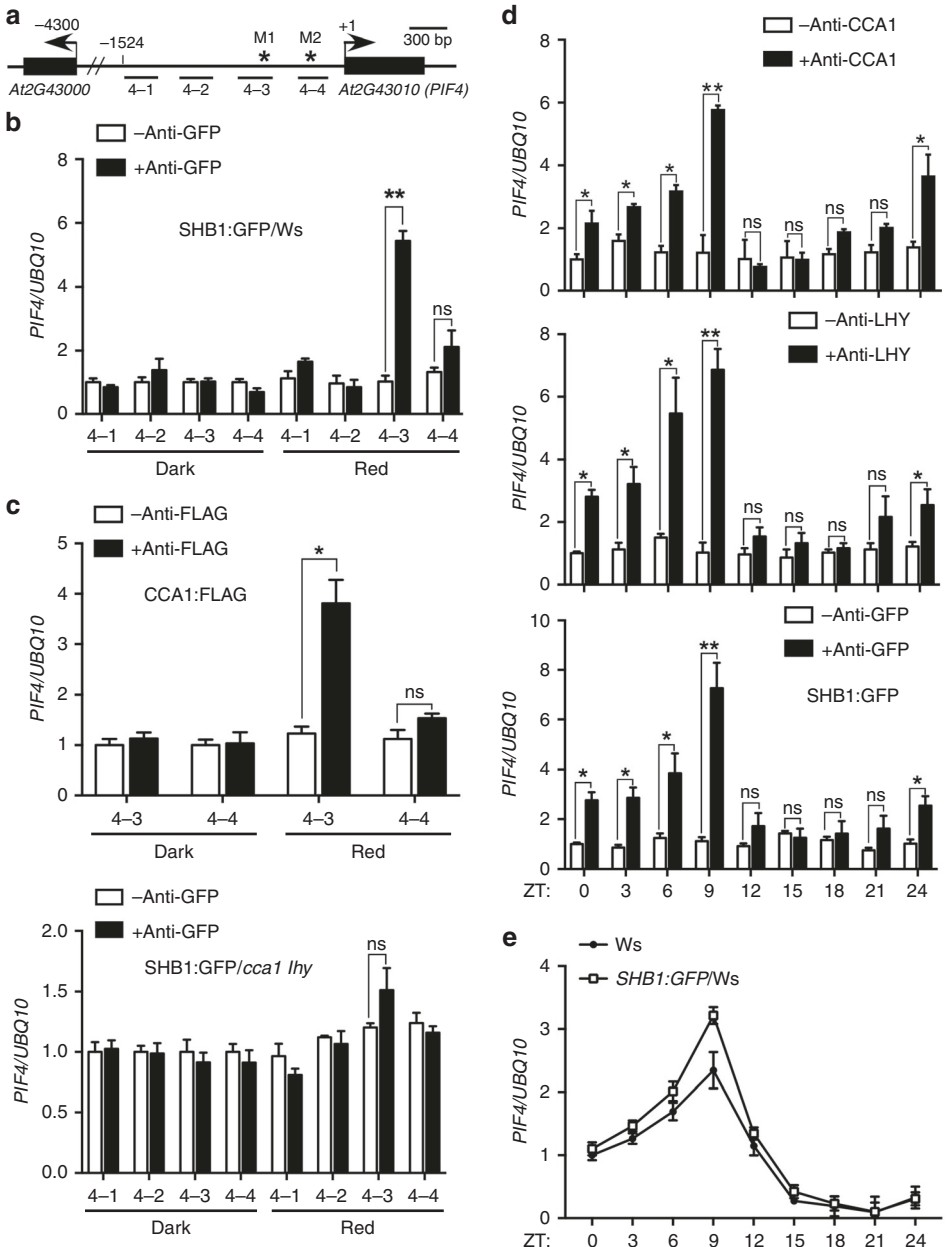

**Fig. 4** SHB1 and CCA1/LHY are associated with the *PIF4* promoter. **a** *PIF4* promoter demonstrating the chromatin regions (4–1, 4–2, 4–3, and 4–4) used for ChIP-quantitative PCR analysis and the potential Myb-binding sites M1 and M2 (asterisks). Association of SHB1:GFP **b** and CCA1:FLAG or SHB1:GFP in *cca1 lhy* **c** with the *PIF4* promoter. **d** Rhythmic association of CCA1, LHY, or SHB1:GFP with the *PIF4* promoter under 12-hr dark and 12-hr red light cycles from two biological replicates. Enrichment of DNA fragments was quantified by qPCR and normalized to that of *UBQ10*. **e** Rhythmic *PIF4* expression at various ZT points in Ws and *35 S::SHB1:GFP* transgenic plants under 12-hr dark and 12-hr red light cycles. Data are presented as the means ± SE calculated from two biological replicates. Source data are provided as a Source Data file

interacts with ELF4. ELF4:GFP was associated with both 4–3 (*p* < 0.018) and 4–4 (*p* < 0.013) fragments in the dark, but less efficiently with the 4–4 (*p* < 0.045) fragment under red light (Supplementary Fig. 5a). ELF4:GFP accumulated at comparable levels in the dark and under red light (Supplementary Fig. 5b). Through ChIP-quantitative PCR assays and genome-wide expression profiling, PRR5, PRR7, and PRR9 also bind to the upstream regions of *PIF4* and other key transcription factor genes, and repress their expression[47]. We hypothesize that the occupation of the *PIF4* promoter by the entire EC complex and/or the PRR proteins in the dark may interfere with the binding of CCA1 or LHY to the *PIF4* promoter.

To correlate SHB1 binding to the *PIF4* promoter with the regulation of *PIF4* expression, we performed rhythmic ChIP assay for CCA1, LHY, and SHB1 (Fig. 4d). We used wild-type plants with antibodies against CCA1 or LHY and *35 S::SHB1: GFP* transgenic plants. We used the 35 S promoter to drive *SHB1* expression given that SHB1 native promoter is very weak and SHB1 expression is not rhythmic. The plants were grown under 12-hr red light and 12-hr dark and were sampled at ZT0 and every 3 h thereafter for 24 h. CCA1, LHY, and SHB1 were associated with the *PIF4* promoter from ZT0 to ZT9 but not thereafter (Fig. 4d). The ChIP peak at ZT9 coincided with a maximum level of *PIF4* expression, and *PIF4* mRNA levels

declined after ZT9 (Fig. 4e). Surprisingly, the CCA1 ChIP peak did not coincide with the *CCA1* mRNA peak (Supplementary Fig. 1c and Fig. 4d). There might be a gradual departure of the EC complex or PRRs. Alternatively, CCA1 and LHY proteins or the SHB1–CCA1/LHY complex is very stable under red light. As shown in an early study, a large number of CCA1 target genes of CCA exhibit a peak expression from ZT5 to ZT16 in LD diel conditions or LL constant light[48]. Among them, *ERD7* has a similar expression pattern as *PIF4*. Therefore, CCA1 plays a potentially large role in the regulation of morning-expressed genes.

**CCA1 and LHY bind the MYB element in the *PIF4* promoter.** We identified two MYB-binding elements, M1 and M2, in the *PIF4* promoter (Fig. 5a and Supplementary Fig. 6a). In a yeast one-hybrid (Y1H) assay, CCA1 and LHY bound to a DNA fragment that contains both M1 and M2 MYB binding elements in the *PIF4* promoter (Fig. 5b). We then tested which MYB-binding element is recognized by CCA1 or LHY using a trimeric M1 or M2 element in tandem repeats (Fig. 5c and Supplementary Fig. 6b). CCA1 and LHY bound to the trimeric M1 element but not the trimeric M2 element. We subsequently mutated the M1 element or M2 element by changing two conserved bases and assessed whether CCA1 or LHY recognizes the mutated element (Fig. 5c and Supplementary Fig. 6b). Neither CCA1 nor LHY bound to the mutated trimeric M1 element.

*PIF4* gene driven by a *PIF4* promoter that bears either a wide type or a deleted M1 box was subsequently introduced into *pif4* or *shb1-D pif4* background (Fig. 5a). Three lines in each category with comparable transgene expression was examined for light- and thermo-responses (Supplementary Fig. 7a, b). The *pPIF4::PIF4* cassette with a deleted M1 box hardly rescued the *pif4* hypocotyl phenotype in either *pif4* or *shb1-D pif4* background compared with that with a wild-type M1 box (Fig. 5d, e and Supplementary Fig. 7c, d). Deletion of the M1 box completely abolished the red light-induced *PIF4* expression (Fig. 5f, g). Given the effect of M1 box deletion on *pif4* hypocotyl phenotype was less-dramatic than that on *PIF4* expression, other signaling pathways may be operated to compensate for the lack of PIF4-mediated hypocotyl elongation in those *pPIF4md::PIF4* lines. Delivery of the wild-type *pPIF4::PIF4* cassette to *pif4* rescued *pif4* hypocotyl thermo-response (Fig. 5h, i and Supplementary Fig. 7e, f). The *pPIF4::PIF4* cassette with a deleted M1 box in either *pif4* or *shb1-D pif4* showed a considerable defect in rescuing *pif4* hypocotyl thermo-response but less-dramatic compared with red light-mediated hypocotyl elongation (Fig. 5h, i and Supplementary Fig. 7e, f). The same cassette with a deleted M1 box in either *pif4* or *shb1-D pif4* also reduced the magnitude of thermo-induced *PIF4* expression but did not alter the thermo-induction pattern compared with the wild type cassette (Fig. 5j, k). Therefore, regulated *PIF4* expression by SHB1–CCA1/LHY only contributes partially to PIF4-mediated thermo-responses.

Either CCA1 or LHY did not bind in vivo the *PIF4* promoter with a deleted M1 box but showed certain affinity toward the base-substituted M1 box (Supplementary Fig. 7g and Supplementary Fig. 8a). In contrast to the in vivo ChIP assays, the Y1H assays may mask the weak affinity of CCA1 or LHY toward the base-substituted M1 box since 300 ng/ml AbA was added in order to titrate off the background growth (Fig. 5b, c). As the base-substituted M1 box was still partially active in vivo, the disruption of the PIF4-mediated light- and thermo-responses by base substitution was less dramatic compared with M1 box deletion (Supplementary Fig. 7g and Supplementary Fig. 8a–j).

**CCA1 and LHY interact with SHB1.** SHB1 and CCA1 or LHY were associated with the *PIF4* promoter, and CCA1 and LHY were required for the association of SHB1 with the *PIF4* promoter. We performed bimolecular fluorescence complementation (BiFC) assays in *N. benthamiana* leaves to test whether SHB1 physically interacts with CCA1 or LHY (Fig. 6a, b). In this system, CCA1 or LHY was fused with YFP$^N$, and SHB1 was fused with YFP$^C$. All constructs were driven by the CaMV 35 S promoter. As controls, CCA1:YFP$^N$ or LHY:YFP$^N$ did not form a fluorescence pair with YFP$^C$, and SHB1:YFP$^C$ did not form a fluorescence pair with YFP$^N$ (Fig. 6a). Both CCA1:YFP$^N$ and LHY:YFP$^N$ interacted with SHB1:YFP$^C$ as shown in fluorescence images. We also presented confocal images for the interaction between CCA1:YFP$^N$ or LHY:YFP$^N$ and SHB1:YFP$^C$ (Fig. 6b).

We next performed co-immunoprecipitation (co-IP) assays with protein extracts prepared from *35 S::SHB1:GFP* and *35 S:: CCA1:FLAG* or *35 S::LHY:MYC* transgenic *Arabidopsis* plants (Fig. 6c). SHB1:GFP coprecipitated CCA1:FLAG or LHY:MYC, further confirming their direct physical interaction. The additional bands recognized by anti-GFP antibodies in the SHB1:GFP lane may represent degradation products of full-length SHB1: GFP (Source Data file). We also detected SHB1 interaction with either CCA1 or LHY by BiFC assays in the dark; however, the interaction occurred less frequently (Supplementary Fig. 9a). We performed Co-IP experiments with the *35 S::SHB1:GFP* transgenic plants in the dark and under red light, and detected CCA1 or LHY using anti-CCA1 or anti-LHY antibodies (Supplementary Fig. 9b). Equal amounts of CCA1 or LHY were precipitated by SHB1:GFP in the dark and under red light.

**SHB1 N terminus interacts with the CCA1 or LHY C terminus.** SHB1 contains an N-terminal SPX domain and a C-terminal EXS domain[38]. The SPX domain retains the function of full-length SHB1, whereas the function of the EXS domain remains unknown. SHB1 localizes to the nucleus, and the EXS domain in SHB1 may exhibit a function distinct from those in other SYG1-like proteins. To map the interaction domain of SHB1 with CCA1 or LHY, we split SHB1 into the SHB1 N terminus (N520) that contains 520 amino acids and the putative SPX domain and the SHB1 C terminus (C325) that contains 325 amino acids and the EXS domain (Supplementary Fig. 10a). Truncated SHB1 N520 or C325 was fused to YFP$^C$ and used in BiFC assays against full-length CCA1 or LHY that was fused to YFP$^N$. The N520:YFP$^C$ fusion protein formed fluorescence pairs with either CCA1:YFP$^N$ or LHY:YFP$^N$ in *Nicotiana* leaves (Supplementary Fig. 10b). The C325: YFP$^C$ fusion protein that contains the EXS domain did not interact with CCA1:YFP$^N$ or LHY:YFP$^N$ in the BiFC assays. Both N520:YFP$^C$ and C325: YFP$^C$ fusion proteins were probed with anti-HA antibodies and expressed at comparable levels (Supplementary Fig. 12a).

CCA1 and LHY are MYB transcription factors and contain a conserved MYB DNA-binding domain at their N termini that is approximately located from amino acids 22–72. We made constructs in which the N-terminal 173 amino acids of CCA1 or LHY were fused to YFP$^N$ (N173:YFP$^N$) or the C-terminal sequence after amino acid 173 was fused to YFP$^N$ (CCA1 C435: YFP$^N$ or LHY C487:YFP$^N$) (Supplementary Fig. 11a). The C terminus but not the N-terminal MYB binding domains of CCA1 or LHY interacted with full-length SHB1 in the BiFC assays (Supplementary Fig. 11b). Various controls were also included to support the specificity of the interactions. Of the total proteins probed with anti-Myc antibodies, CCA1 or LHY N173:YFP$^N$, CCA1 C435:YFP$^N$ and LHY C487:YFP$^N$ fusion proteins were expressed at comparable levels (Supplementary Fig. 12b).

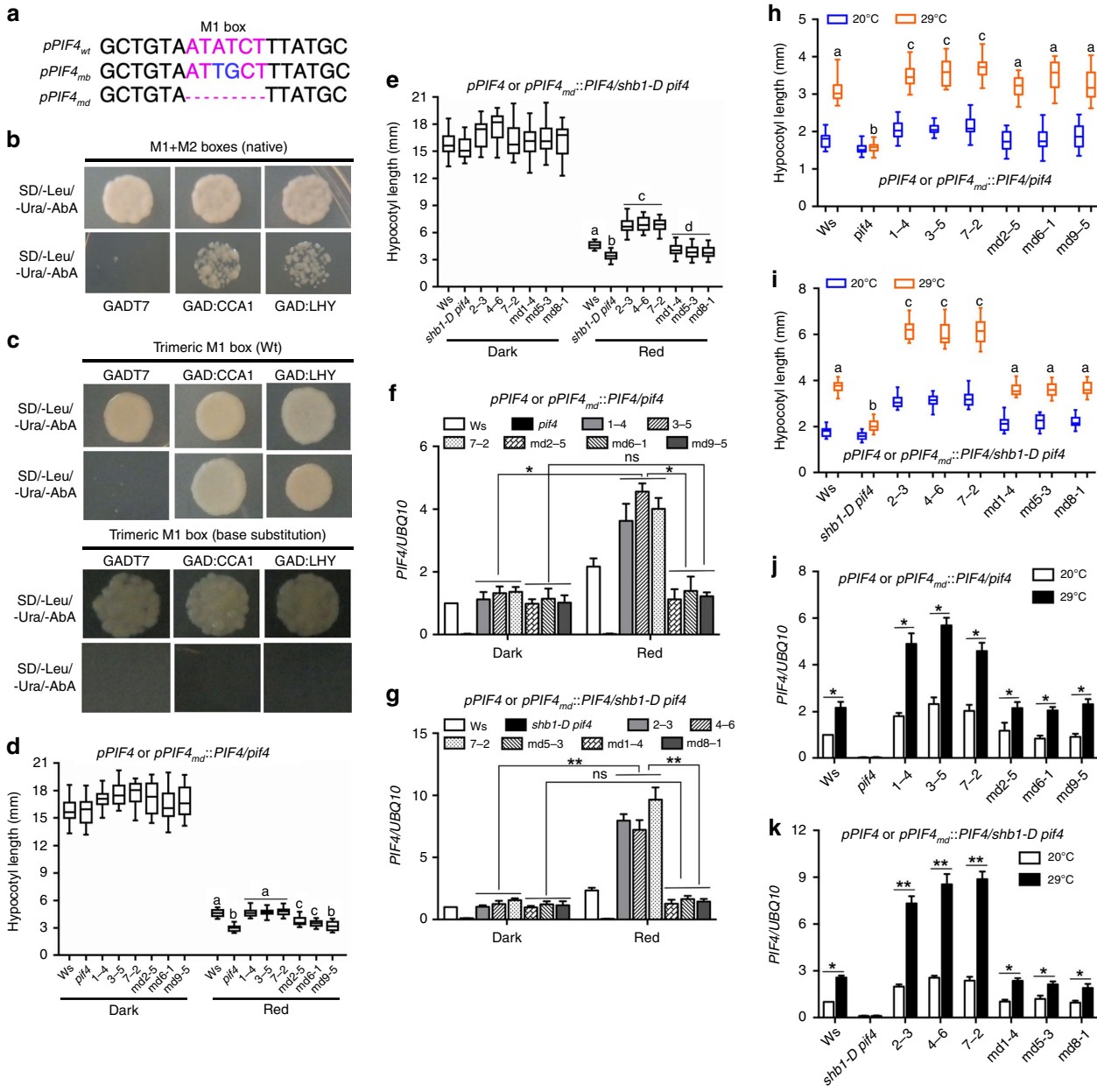

**Fig. 5** CCA1 and LHY recognize MYB M1 box in the *PIF4* promoter. **a** Potential MYB-binding element M1 is noted in red, and the bases mutated are noted in blue. mb and md indicate mutation by base substitution and deletion, respectively. Yeast one-hybrid assays for CCA1 or LHY in *pGADT7* over a 424-bp fragment that contains both M1 and M2 elements from the *PIF4* promoter in the *pAbAi* vector **b** and for CCA1 or LHY in *pGADT7* over trimeric repeats of wild type (upper) or mutated (lower) M1 element **c**. Golden Y1H cells were grown on SD/-Leu/-Ura media with or without 300 ng/ml Aureobasidin A (AbA) selection. Hypocotyl lengths of Ws, *pif4* and *pPIF4* or *pPIF4md::PIF4* in *pif4* **d** and Ws, *shb1-D pif4* and *pPIF4* or *pPIF4md::PIF4* in *shb1-D pif4* **e**. *PIF4* expression in Ws, *pif4* and *pPIF4* or *pPIF4md::PIF4* in *pif4* **f** and Ws, *shb1-D pif4* and *pPIF4* or *pPIF4md::PIF4* in *shb1-D pif4* **g**. The seedlings were grown in the dark or under 10 μmol m$^{-2}$ s$^{-1}$ red light for 4 days. Hypocotyl lengths of Ws, *pif4* and *pPIF4* or *pPIF4md::PIF4* in *pif4* **h** and Ws, *shb1-D pif4* and *pPIF4* or *pPIF4md::PIF4* in *shb1-D pif4* **i** under 30 μmol m$^{-2}$ s$^{-1}$ white light at 20 °C for 7 days or 20 °C for 4 days followed by 29 °C for 3 days. *PIF4* expression in Ws, *pif4* and *pPIF4* or *pPIF4md::PIF4* in *pif4* **j** and Ws, *shb1-D pif4* and *pPIF4* or *pPIF4md::PIF4* in *shb1-D pif4* **k** grown at 20 °C for 5 days and then incubated at 20 °C or 29 °C for 4 h. Significance levels by Student's two-tailed heteroscedastic *t* tests in **d**: $p < 0.001$ between **a** and **b**, $p < 0.01$ between **a** and **c**, $p < 0.05$ between **b** and **c**; in **e**: $p < 0.001$ between **a** and **b**, **a** and **c**, **b** and **c** or **c** and **d**, $p < 0.01$ between **a** and **d**, $p < 0.05$ between **b** and **d**; in **h**: $p < 0.001$ between **a** and **b** or **b** and **c**, $p < 0.01$ between **a** and **c**; in **i**: $p < 0.001$ between **a** and **b**, **a** and **c** or **b** and **c**. Source data are provided as a Source Data file

**CCA1 and LHY co-act with SHB1.** SHB1 and CCA1 or LHY were required for *PIF4* expression giving that knocking down of *SHB1* or knocking out of *CCA1/LHY* reduced *PIF4* expression (Fig. 1a, e). We conducted in vivo trans-activation experiments in *Arabidopsis cca1 lhy* protoplasts with various constructs (Fig. 6d). We used a *PIF4* promoter::LUC reporter along with a CaMV 35 S::REN residing on the same vector to control

differences in transformation efficiency. We delivered CCA1 and LHY driven by their native promoter and SHB1 driven by 35 S promoter as effectors to *cca1 lhy* (Fig. 6d). We used the 35 S promoter for SHB1 giving that its native promoter drives a very low level of expression and SHB1 is not rhythmically expressed (Supplementary Fig. 1c). Compared with controls without any effector proteins delivered, delivery of the *pCCA1::*

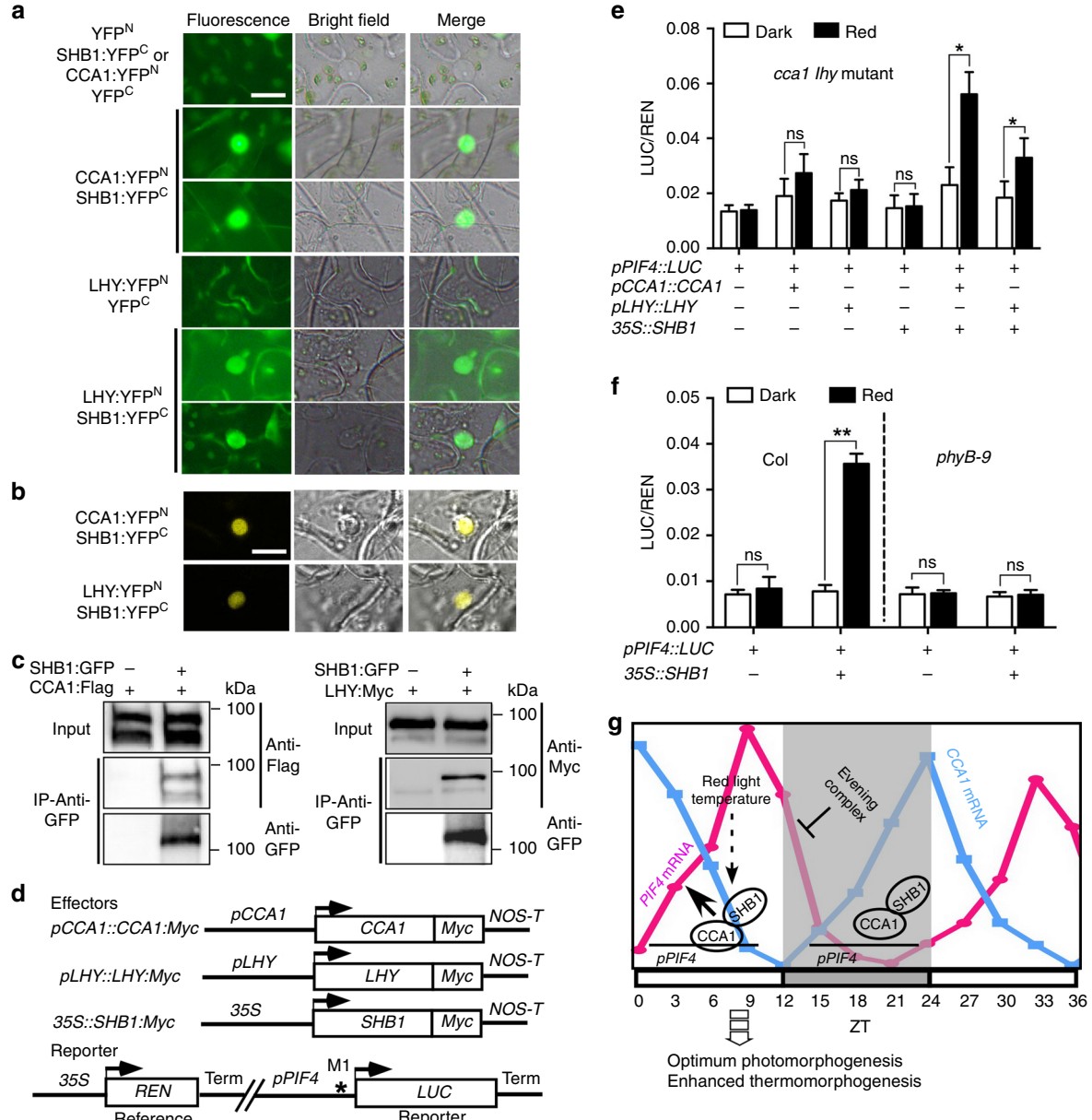

**Fig. 6** SHB1 co-acts with CCA1 and LHY. **a** BiFC fluorescent images of YFP$^N$ with SHB1:YFP$^C$, CCA1:YFP$^N$ or LHY:YFP$^N$ with YFP$^C$, and CCA1:YFP$^N$ or LHY: YFP$^N$ with SHB1:YFP$^C$. **b** BiFC confocal images of CCA1:YFP$^N$ or LHY:YFP$^N$ with SHB1:YFP$^C$. Images were captured 48–72 h in *Nicotiana* leaf epidermal cells after Agrobacteria transfection. Bar = 10 μm. **c** Co-immunoprecipitation (Co-IP) of CCA1:FLAG or LHY:MYC by SHB1:GFP in protein extracts prepared from transgenic *Arabidopsis* with anti-GFP antibody. **d** Effector, reporter, and reference constructs used. Arrowheads indicate transcription start sites, and NOS-T represents polyadenylation signal from the nopaline synthase gene. Asterisk indicates the location of the M1 element in the *PIF4* promoter. Relative LUC/ REN activity in *cca1 lhy* transformed with a reporter along with *pCCA1::CCA1:MYC*, *pLHY::LHY:MYC* or *35 S::SHB1:MYC* **e** and in Col or *phyB-9* transformed with reporter in the absence or presence of *35 S::SHB1:MYC* **f** in the dark or under red light. Source data are provided as a Source Data file. **g** A model explaining the regulation of CCA1 or LHY and SHB1 over *PIF4* expression. *PIF4* expression is repressed by the evening complex (EC) or PRRs at early evening. Toward the end of night and after dawn, CCA1 recruits SHB1 to mediate red light induction of *PIF4* expression. This constitutes a signaling loop to optimize photomorphogenesis and enhance thermomorphogenesis

*CCA1:MYC* or *pLHY::LHY:MYC* construct alone to *cca1 lhy* barely activated the expression of *LUC* from the *PIF4* promoter either in the dark or under red light (Fig. 6e). Delivery of the *35 S::SHB1:MYC* construct alone to *cca1 lhy* also exhibited minimal effects on the transcription of *LUC* from the *PIF4* promoter.

When *pCCA1::CCA1:MYC* and *35 S::SHB1:MYC* were codelivered to *cca1 lhy*, the transcription of *LUC* from the *PIF4* promoter was activated weakly in the dark but strongly under red light (Fig. 6e). The effects on *LUC* transcription from the

*PIF4* promoter were considerably less dramatic when *35 S:: SHB1:MYC* was codelivered with *pLHY::LHY:MYC* either in the dark or under red light. CCA1 protein levels were comparable to that of LHY protein (Supplementary Fig. 12c). CCA1 or LHY likely recruits SHB1 to the *PIF4* promoter, and SHB1 mediates red light-induced *PIF4* expression. This red light induction was blocked in *phyB-9* under red light (Fig. 6f). The levels of SHB1, CCA1, and LHY proteins were not different between transformed protoplasts maintained in the dark or under red light (Supplementary Fig. 12c). The nuclear localization of SHB1:

GFP was not affected in the dark compared with that under red light (Supplementary Fig. 12d).

## Discussion

Light and temperature are two important environmental signals that regulate plant growth and development. However, excess light often damages the photosynthetic apparatus and frequently causes photoinhibition. Plants thus evolve a series of strategies to achieve optimum photomorphogenesis. Here we demonstrate a regulatory desensitization step involving SHB1, CCA1/LHY, and PIF4. We proposed a model explaining how CCA1/LHY and SHB1 upregulate *PIF4* expression (Fig. 6g). Our study suggests that the highly expressed circadian clock proteins CCA1 and LHY in the morning recruits SHB1 to activate red light-induced *PIF4* expression and desensitize light responses. PIF4 is also a positive regulator of thermomorphogenesis. When temperature increases after morning, the interaction of SHB1 with CCA1 may sustain *PIF4* expression to allow plants to better adapt to the temperature-increasing environment. Alternatively, *PIF4* transcription allows plants to monitor day-time shade and temperature, which eventually influences PIF4 protein accumulation and activity.

The EC formed by LUX, ELF3, and ELF4 proteins peaks at dusk, binds to the *PIF4* promoter via the LUX transcription factor, and represses *PIF4* transcription in the early evening[46]. EC may recognize two binding elements near the CCA1 and LHY-binding elements. The occupation of the elements by EC and possibly PRRs in the dark may interfere with the binding of CCA1 and LHY to the *PIF4* promoter. Indeed, CCA1 and LHY were not associated with the *PIF4* promoter in the dark (Fig. 4c and Supplementary Fig. 4c). Near dawn, CCA1 and LHY associate with the *PIF4* promoter, recruit SHB1 to activate *PIF4* transcription and promote growth, such as hypocotyl elongation (Fig. 6g). CCA1 was originally discovered as an activator of a dawn gene LHCB[12]. CCA1 and LHY also repress the evening element-containing genes at dawn[49]. We discovered a new role of CCA1 and LHY to directly recruit SHB1 that mediates red light-induced expression of *PIF4* in the morning and thereafter. As such, *PIF4* transcription peaks after the middle of the day.

ChIP-seq analysis identified the regions of CCA1 occupancy at many EE-containing, evening-expressed, clock-regulated genes in the *Arabidopsis* genome[48]. CCA1 recognition elements are also located near many genes with peak expression in the morning and in proximity to genes that do not cycle in LL conditions. CCA1 recognizes the canonical EE (AAATATCT) in LL conditions, and the binding capacity of CCA1 is more affected by alteration in the TCT sequence compared with alterations in the ATA or AA[50]. In our yeast one-hybrid analysis, CCA1 only recognizes the M1 element sequence ATATCT, not the M2 element sequence ATATCA in the *PIF4* promoter (Fig. 5a, c and Supplementary Fig. 6). The last base in the core sequence is apparently critical for CCA1 recognition. In addition, the flanking sequences may also affect the affinity of CCA1 to its binding elements.

PIFs belong to a large family of basic helix-loop-helix (bHLH) proteins and exhibit a variety of different functions[51]. Monogenic *pif3*, *pif4*, *pif5*, and *pif7* null mutants exhibit light-hypersensitive seedling phenotypes, whereas *PIF4*, *PIF5*, and *PIF7* are clock regulated[8,52]. The three PIFs promote the elongation of hypocotyls and petioles during vegetative growth in a redundant manner[39,53–55]. PIF4 has a more-prominent role than PIF5[56,57]. Among the *PIF* genes, SHB1 specifically regulates the red light-induced expression of *PIF4* and likely *PIF7* (Fig. 1a and Supplementary Fig. 1a). Light-activated phyB interacts with PIF4 and targets it for post-transcriptional degradation following the entire light period[53,58,59]. In these experiments, a CaMV 35 S promoter-driven *PIF4* construct was used. On the other hand, PIF4 reaccumulates after a few hours of light treatment[60,61]. When *PIF4* was driven by its native promoter in a recent study, PIF4 protein accumulated in the light period from ZT0 to ZT8 but not in the dark period from ZT12 to ZT20[42]. Despite continuous PIF4 degradation, the upregulation of *PIF4* expression by CCA1 and SHB1 throughout the day might be responsible for the reappearance of PIF4 protein, constituting a negative regulatory step to antagonize photomorphogenesis.

## Methods

**Plant materials and growth conditions**. *Arabidopsis thaliana* ecotype Columbia-0 (Col) and Wassilewskija (Ws) were used as wild-type plants. The mutant lines *shb1*, *shb1-D*, and *pif4* were described previously[37,53]. *cca1–11*, *cca1–11 lhy-21*, and *ELF4:GFP* seeds were from Gang Li, and *lhy-21* seeds were from ABRC (https://abrc.osu.edu/). The *shb1-D pif4* double mutant was generated by crossing *shb1-D* to *pif4* and PCR-genotyped using the primers described previously[37]. *shb1 cca1 lhy* or *shb1-D cca1 lhy* triple mutant was generated by crossing *shb1* or *shb1-D* to *cca1 lhy* double mutant and PCR-genotyped using the primers listed in table S1. All plants were grown in a growth room with a 16 L/8D cycle at 22 °C for seed propagation. For circadian expression analysis, *Arabidopsis* seedlings were entrained under a 12-hr white or red light and 12-hr dark cycle for 7 days, and then released to continuous white or red light for the following days as described in the figure legends. Similar results were observed when seedlings were released to continuous white or red light for one day and then sampled for the following days.

**Quantitative RT-PCR**. For quantitative reverse transcription-polymerase chain reaction (RT-PCR) analysis, total RNA was extracted from seedlings using the MiniBEST Plant RNA Extraction Kit (Takara) or the SV Total RNA Isolation Kit (Promega). SuperScript™ II Reverse Transcriptase (Invitrogen) was used to synthesize cDNA from the RNA. Quantitative real-time PCR was performed with the TransStart Tip Green qPCR SuperMix (Transgen Biotech) on a QuantStudio™ 6 Flex Real-Time PCR machine. The thermal cycling program was 95 °C for 5 min, followed by 40 cycles of 95 °C for 10 s, 56 °C for 10 s and 72 °C for 20 s. The last step involves a one-cycle dissociation stage at 95 °C for 5 s, 65 °C for 1 min, 98 °C for 1 min and 40 °C for 10 s. Most RT-PCR were performed with three biological replicates unless it is specifically indicated in the figure legends. Each biological replicate was represented by three technical replicates. The expression levels of specific genes were normalized to that of *UBQ10* and were presented relative to the expression levels in wild type. The gene-specific primers for qRT-PCR analysis are listed in Supplementary Table 1.

**Hypocotyl length measurement**. Seeds were sterilized by 1% (v/v) sodium hypochloride and 0.2% (v/v) sodium dodecyl sulphate (SDS) and plated on ½ MS medium. After a 3-day vernalization at 4 °C, seeds were treated with white light for 2–3 h and then incubated under 10 µmol m$^{-2}$ s$^{-1}$ red light for 4 days. For thermomorphogenesis, *Arabidopsis* seedlings were grown under continuous fluorescent white light (30 µmol m$^{-2}$ s$^{-1}$) at 20 °C for 7 days or at 20 °C for 4 days followed by growth at 29 °C for 3 days. More than 50 seedlings were photoimaged and hypocotyl lengths were measured using ImageJ software (http://rsb.info.nih.gov/ij). Hypocotyl experiments were repeated thrice, and one representative dataset was shown. About 50 seedlings were measured for each genotype. For *PIF4* expression analysis at 29 °C, seedlings were grown at 20 °C for 5 days and then incubated at either 20 °C or 29 °C for 4 hr before total RNA was extracted.

**Plasmid construction**. All primers used to make plasmid constructs were listed in Supplementary Table 1. All constructs, except for the plasmids used for yeast one-hybrid (Y1H) assay, were made by using the gateway system. In general, the full-length genomic coding region was PCR amplified from Col genomic DNA, cloned into the pCR8/GW/TOPO TA vector, and then recombined into the corresponding destination vectors. To examine the in vivo function of M1 element in *PIF4* promoter, a genomic fragment that contains *PIF4* promoter up to −1524 bp and coding sequence was cloned into the TA vector. This TA clone was then used as template to mutate the M1 element as base substitution or deletion by a reverse-PCR procedure (TOYOBO, SMK-101). Transgenes were verified by using PIF4-F and c-Myc-R primers.

**Protein isolation and western blot**. To detect rhythmic PIF4 accumulation in Col, *shb1*, Ws, and *shb1-D*, seedlings were first grown under 12-hr 2 µmol m$^{-2}$ s$^{-1}$ red light and 12-hr dark for 5 days and then sampled at ZT0 and every 3 h thereafter for 24 h. Protein isolation buffer contains 100 mM MOPS pH 7.6, 100 mM NaCl, 10% glycerol, 40 mM 2-mercaptoethanol, 5% SDS, 1× protease inhibitor cocktail from Roche, and 2 mM phenylmethylsulfonyl fluoride (PMSF). Eighty µl buffer were added to 100 µg grinded powder. The mixture was immediately heated at 70 °C for 10 min and separated on 8% SDS–PAGE gel. PIF4 protein was monitored

by western blot using anti-PIF4 antibody AS16 3955 (Agrisera, Sweden) from three biological replicates. Protein quantification was performed with IMAGEJ normalized to an actin band.

To examine the accumulation of tagged proteins in transgenic plants, the proteins were isolated in buffer that contained 50 mM Tris-HCl pH 8.0, 10 mM EDTA, 1% SDS, 1 mM PMSF and 1× protease inhibitors (Roche). The tagged proteins were detected in western blot using anti-GFP ab1218 (Abcam, Cambridge, MA), anti-FLAG ab49763 (Abcam, Cambridge, MA) and anti-MYC ab32 (Abcam, Cambridge, MA) antibodies. To examine the accumulation of tagged proteins in transient assays, the proteins were isolated in buffer that contained 50 mM Tris-HCl pH 7.5, 150 mM NaCl, 0.1% NP-40, 4 M urea, and 1 mM PMSF. The tagged proteins were detected by anti-HA CW0092 (CWBIO, China) and anti-MYC antibodies as described above. Total proteins in Supplementary Fig. 12c were extracted from protoplasts by boiling in 2× SDS loading buffer at 95 °C for 10 min. The tagged proteins were detected using anti-GFP and anti-MYC antibodies as described above.

**ChIP assay**. ChIP assays were performed as described previously[44,62]. Seedlings that have *35 S::SHB1:GFP*, *35 S::CCA1:FLAG* or *35 S::LHY:MYC* transgene were grown in the dark or under 15 μmol m$^{-2}$ s$^{-1}$ red light for 5 days, and then cross-linked for 30 min in 1% formaldehyde solution under vacuum. For rhythmic ChIP analysis, Col wild type and *35 S::SHB1:GFP* transgenic plants were grown under 12-hr red light and 12-hr dark for 3-weeks, and sampled at ZT0 and every 3 h thereafter for 24 h. The plant materials were ground for 10 min on ice in nuclear isolation buffer with 1% formaldehyde, and cross-linking was quenched by adding glycine to a final concentration of 125 mM. The chromatin complex was isolated by nuclear lysis buffer (50 mM Tris-HCl pH 8.0, 10 mM EDTA, 0.5% SDS, 0.1 mM PMSF and 1 × protease inhibitors) and sheared by sonication to generate fragments that were ~ 300–500 bp. The sonicated chromatin complex was then immuno-precipitated by anti-GFP, anti-FLAG, and anti-MYC antibodies as previously described. For rhythmic ChIP analysis, the sonicated chromatin complex was harvested by anti-CCA1 R1234–3 (Abiocode, Agoura Hills, CA), anti-LHY R3095–2 (Abiocode, Agoura Hills, CA) and anti-GFP antibodies. After the reversal of cross-linking, DNA was precipitated in the presence of glycogen (Thermo Fisher) and analyzed by ChIP-qPCR. Most ChIP-PCR were performed with three biological replicates unless it is specifically indicated in the figure legends. Each biological replicate was represented by three technical replicates. The fold enrichment of the specific chromatin fragment was normalized to the *UBQ10* amplicon. All primers used for ChIP-qPCR were listed in Supplementary Table 1.

**Yeast one-hybrid (Y1H) assay**. A native 424-bp DNA fragment in the *PIF4* promoter was PCR amplified and cloned into the Hind III and Sal I sites of the *pAbAi* vector (Clontech). Three tandem copies of the putative CCA1/LHY-binding element M1 or M2 were synthesized as oligonucleotides and ligated into the Hind III and Sal I sites of the *pAbAi* vector (Clontech). Each repeat of the element has a GCTGTA**ATATCT**TTATGC or TTCCAC**ATATCAG**GTTAT sequence with a 6-bp flanking sequence on either side of the core elements. The *pAbAi* vectors harboring the constructs were integrated into the genome of the Y1H Gold yeast strain. The coding sequences of *CCA1* and *LHY* were PCR amplified from cDNA generated from *Arabidopsis* Col total RNA and cloned into the pGADT7-AD vector (Yeast Protocols Handbook by Clontech, Mountain View, CA). Transformants were selected on minimal synthetic dropout (SD) medium lacking Leu, Trp, and Ura. Yeast cells grown in SD/-Leu/-Trp/-Ura broth were diluted to OD$_{600}$ 0.5 and plated on SD/-Leu/-Trp/-Ura plate with or without AbA as described by the Matchmaker Gold Yeast One-Hybrid Library Screening System Protocol (Clontech, Mountain View, CA).

**BiFC assay**. BiFC experiments were conducted as described[63]. To generate the BiFC constructs, *CCA1* or *LHY* cDNA and *SHB1* genomic sequence without their stop codons were PCR amplified and subcloned into the binary vectors *pSPYNE* and *pSPYCE* under the control of the cauliflower mosaic virus (CaMV) 35 S promoter. pSPYNE::CCA1 or LHY was co-transformed with pSPYCE::SHB1 into *N. benthamiana* leaves by agroinfiltration. Their interactions in *N. benthamiana* leaves were analyzed using an Olympus fluorescence microscope (Olympus BX53 with a DP26 CCD camera) or a confocal microscope (Leica TCS SP5 II).

**Co-IP assay**. For co-IP assays, total protein of *35 S::SHB1:GFP* and *35 S::CCA1: FALG* or *35 S::LHY:MYC* seedlings was extracted using IP buffer (10% Glycerol, 25 mM Tris-HCl pH 7.5, 1 mM EDTA, 150 mM NaCl, 0.1% Triton-100, 2 mM β-mercaptoethanol, 1 mM PMSF and 1 × protease inhibitors). After centrifugation at 13,000 rpm for 20 min, the two sources of proteins were mixed and incubated for 2–3 h at 4 °C. The mixture was then incubated with the anti-GFP rabbit polyclonal antibody 10004D (Thermo Fisher, CA) for another 2 h. The beads were washed four times with IP buffer. The pelleted beads were boiled in 60 μl 2 × SDS buffer and separated by 8% SDS–PAGE. Western blots were probed by using anti-FLAG or anti-MYC antibodies at a 1:1000 dilution.

**Transient trans-activation assay**. *Arabidopsis* mesophyll protoplast isolation and transformation were performed as described[64]. *pCCA1::CCA1:MYC*, *pLHY::LHY:*

*MYC* and *35 S::SHB1:MYC* were used in effector constructs. The *pGreen II−0800-LUC* vector that bears *pPIF4::LUC* and *35 S::REN* was used as a reporter construct. Luciferase activity was measured using the Dual-Luciferase Reporter Assay System (Promega, USA). Relative luciferase activity was normalized to REN activity as a LUC/REN ratio for each biological sample. Experiments were repeated thrice and each biological replicate was represented by three technical replicates.

## Data availability

The authors declare that all data supporting the findings of this study are available within the paper and its supplementary information files. The source data underlying Figs. 1, 2, 3a–d, f, 4b–e, 5d–k, and 6c, e, f and Supplementary Figs. 1a–c, 2a–c, g, 4a–f, 5a, b, 7a, b, g, 8b–j, 9b, and 12a–c are provided as a Source Data file.

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

## Acknowledgements

We thank Gang Li for *35 S::CCA1:FLAG*, *cca1-11*, *cca1-11 lhy-21*, and *35 S::ELF4:GFP* seeds and ABRC for *lhy-21* seeds. This work was supported by a grant from National Natural Science Foundation of China (31470379) and a grant from the USDA National Institution of Food and Agriculture grant 2011–67013–30150.

## Author contributions

M.N. and Q.S. conceived and designed the research plan. Q.S., S.W., G.X., X.K., and M.Z. performed the experiments. Q.S. and M.N. wrote the manuscript.

## Additional information

**Competing interests:** The authors declare no competing interests.

