## [Peer Review File · Nature Communications]

Reviewers' comments:

Reviewer #1 (Remarks to the Author):

The manuscript by Sun et al describes work investigating the role of SHB1 in activating PIF4 through interaction with the clock components CCA1 and LHY. The results provide convincing evidence for SHB1 interaction with CCA1 and thereby recruited to the PIF4 promoter under red light but not in the dark. The author claimed that this mechanism provides a negative regulation of photomorphogenesis, de-sensitization of light response, and enhancement of thermomorphogenesis. However, another way of interpretation is that transcription of PIF4 allows plants to monitor day-time shade and temperature, which alters PIF4 protein accumulation and activity posttranslationally. The findings are of interesting to a broad readership in the fields of plant biology and circadian rhythms. However, the progress is incremental compared to previous study of SHB1 function and circadian regulation of PIF4 expression. Further investigation of the mechanism of light regulation of SHB1 activity, or contribution of SHB1 to circadian regulation of PIF4 expression, would enhance the story. Below are specific comments and suggestions,.

Fig 2a: The *shb1-D* mutation still increased hypocotyl length in the *pif4* mutant background, indicating that SHB1 can promote hypocotyl elongation independent of PIF4. Possibility of addition PIF4-independent mechanism for SHV1 action should be discussed.

Fig 2C: The results are sufficient to show that PIF4-ox can increase hypocotyl elongation independent of CCA1 and LHY (in *cca1/lhy* mutant background). Same 35S-PIF4/*cca1/lhy* lines should be crossed to wild type and *cca1/lhy*, to avoid T-DNA positional effect on PIF4 expression level.

Fig 3 shows that heat activation of PIF4 expression and thermomorphogenesis in the *shb1* and *cca1/lhy* mutant. While the hypocotyl lengths are affected in the mutants, the fold of induction by warm temperature seems unaffected, except for Fig 3a which shows same hypocotyl length between *shb1* and wild type at 22 C (but this is different from figure 2a). Should discuss clearly that SHB1 and CCA1 are not required for thermo-activation of PIF4, but the overall increase of PIF4 level increases the magnitude of heat response. Again, the value of the studied mechanism seems to provide high transcription of PIF4 and hence proper responses to shade and warm temperature, as the PIF4 activity is determined by the combination of transcriptional and posttranslational regulation, while transcription is prerequisite to posttranslational regulation.

Fig 4 shows CCA1/LHY-dependent SHB1 binding to PIF4 promoter in vivo. The results are clear. Why is there no binding in the dark? How does red light induce CCA1/LHY binding to PIF4 promoter? It would be nice to answer these important mechanistic questions.

Fig 5 shows CCA1 and LHY binding to PIF4 promoter element in Y1H assays. It would be nice to show SHB1 binding the DNA only in the presence of CCA1 in such assays (SHB1-AD + CCA1+reporter).

Fig 6 shows BiFC and co-IP data for SHB1-CCA1/LHY interaction in plant cells. Is the interaction affected by darkness and red light?

Fig 7: Reporter gene assays show activation of PIF4 promoter expression by SHB1, CCA1, and combination of the two. The author claimed that CCA1 and LHY act synergistically with SHB1. However, no data was shown for LHY+SHB1 and the data for CCA1+SHB1 shows additive not synergistic (more than the sum of two individually) effect. It's interesting that CCA1 activity is independent of light but that of SHB1 is, and LHY has no activity. Completing this experiment with LHY+SHB1, and testing light effect on CCA1/LHY interaction with SHB1 (Fig 6) could potentially reveal important mechanisms and answer a major outstanding question.

Fig 7c: in the diagram, the curves for CCA1 and PIF4 don't have a 24-hr period. The enhancement of PIF4 transcription by SHB1 is not shown (e.g. by a thicker arrow).

Language can be improved.

Overall, answering any of the following questions could significantly enhance the manuscript: 1. How does red light activate SHB1? Is SHB1 accumulation, nuclear localization, or interaction with CCA1 dependent on phytochrome signaling? 2. How important is SHB1 for CCA1/LHY function beyond PIF4 regulation? How are the circadian profile of PIF4 RNA or other clock-regulated genes (or reporter gene) affected in the *shb1* and *shb1-D* mutants? 3. Is SHB1 important for thermo-adaptation and heat tolerance?

Reviewer #2 (Remarks to the Author):

The manuscript by Sun et al described an intriguing regulation of PIF4 expression by the CCA1-SHB1 protein complex. This study suggested plants have evolved to take advantage of the circadian and timely expression of CCA1 to form a transactivation complex with SHB1 to trigger PIF4 expression after dawn. The up-regulation of PIF4 could antagonize the increasing light signal, as well as respond to the increasing temperature toward noon. In general, the desensitizing model is interesting. However, more experimental data would help to confirm the up-regulation of PIF4 by CCA1-SHB1 complex after dawn. A few specific points are listed below.

1. Statistical assessments should be applied to all figures involving qRT-PCR.

2. The genetic interaction between SHB1 and PIF4. In Fig. 2a, results indicated, and supported by statistical analyses, that *shb1 pif4* double mutant showed intermediate phenotype between *shb1* and *pif4* single mutants. This indicated that the actions of SHB1 do not entirely go through PIF4? Therefore, the conclusion that PIF4 functioning downstream of SHB1 needs to be either toned down or alternative explanations should be provided/discussed.

3. Whether PIF4 functions downstream of CCA1/LHY? In Fig. 2c, the overexpression of PIF4 in *cca1 lhy* double mutants resulted in elongated phenotype. The authors concluded that PIF4 acts downstream of CCA1 and LHY. In fact, based on this data, PIF4 could also function in an independent pathway to promote hypocotyl elongation. It might be helpful to combine the results in Fig. 1 (the R-induced PIF4 relies on CCA1/LHY) in concluding that PIF4 acts downstream of CCA1/LHY.

4. One of the key conclusions in this manuscript is that CCA1 and SHB1 together induce the expression of PIF4 under high temperature, leading to the hypocotyl elongation. However, as shown in Fig. 3b and 3d, hypocotyls of *cca1 lhy* double mutant still elongate in response to high temperature and PIF4 was still upregulated under high temperature in *cca1 lhy*. If the data were shown in fold changes, perhaps the degree of up-regulation was comparable between wild-type and the mutants. This indicated that, under high temperature, the induction of PIF4 does not entirely rely on CCA1/LHY? Similarly results for PIF4 up-regulation and elongated hypocotyl were also observed in *shb1* mutant (Fig. 3a, 3d), indicating the induction of PIF4 under higher temperature is not entirely dependent on SHB1?

5. To claim SHB1 binds to the promoter of PIF4, SHB1-YFP was used. It was, however, unclear whether SHB1:YFP was biologically functional? Was functional complementation performed? Were overexpression lines used? What was the ZT used for ChIP? Also, ChIP should be presented as % input not normalized to UBQ10. For a meaningful comparison in Fig. 4d, the expression level of SHB1 in WT and *cca1 lhy* mutant should be carefully quantitated and used for normalization?

6. The protein-protein interaction data shown in Fig. S6: for those showing no protein-protein interactions, did the constructs yield protein products with comparable expression levels? Also, as shown in Fig. 6 and S6, LHY could interact with SHB1. However, LHY cannot transactivate PIF4 promoter (Fig. 7). Whether the LHY-SHB1 interaction possesses biological significance should at least be discussed.

7. For the transactivation assays shown in Fig. 7b, the transfection of the effector 35S::SHB1

showed R-dependent transactivation activities whereas 35S::CCA1 has comparable transactivation activity toward PIF4::LUC reporter construct under dark and R? Were the SHB1 protein levels different between protoplasts under dark and red light conditions? How about the CCA1 protein levels? The Myc tag for the effector proteins should allow the detection of the effector proteins in the protoplasts. The transactivation activities of different effector proteins could also be normalized by the levels of the effector proteins. For example, is CCA1 indeed much more effective in triggering PIF4 expression, or it is simply CCA1 protein accumulates to a higher level than LHY?

8. A time course binding of CCA1/SHB1 to PIF4 promoter could be examined to better support the model shown in Fig. 7c? In addition, PIF4 protein should accumulate during the day through the action of CCA1-SHB1 complex in triggering PIF4 expression for thermomorphogenic responses. Previous reports showed overexpressed PIF4 was effectively degraded after short period of R light treatment (Kumar et al 2012). On the other hand, Zhang et al 2017 and Lorrain et al 2008 showed that PIF4 re-accumulates after a few hours of light treatment, in line better with the model shown in Fig. 7c. Is CCA1-SHB1 complex responsible for the reappearance of PIF4 protein in the later cases?

Minor points:

1. Fig. S2b and S2c is reversed in order?
2. Line 53-55, TOC1 has been shown to be direct repressors of CCA1. The description should be corrected.
3. In Fig. S1, the red light induced upregulation of PIF7 was also compromised in shb1 mutant. The description in line 120-122 should be slightly modified to reflect the result?
4. Line 158, Fig 2b should be cited.
5. Line 206, despite marked in manuscript text, no data for M2 could be found in Fig. 5c.
6. The band corresponded to SHB1-YFP should be marked in Fig. 6c.
7. Line 234, Fig. S5a should be cited instead of S6a.
8. Line 259, "CCA1 or LHY likely possesses a partial trans-activation activity beside its ability to bind DNA". It was unclear what this statement was referring to?

Reviewer #3 (Remarks to the Author):

In this work, Sun and co-authors identify a novel layer of regulation of PIF4 expression by SHB1 and the clock components CCA1/LHY. Previous evidence had shown other clock components repressing PIF4 expression at evening (Evening Complex and PRRs, the latter not mentioned by the authors). Authors show that both CCA1 and SHB1 bind to the PIF4 promoter to induce its expression in response to red light during deetiolation, and also in response to high temperature in light-grown seedlings. The authors conclude that these factors cooperatively induce PIF4 expression through their direct interaction. Authors also provide genetic evidence showing that this regulatory network inhibits photomorphogenesis and induces thermomorphogenesis.

The question is interesting and novel, and, overall, experiments appear well-performed and data well-analyzed. However, in the current form there is not enough genetic and molecular data to sustain the central conclusions leading to the model presented in figure 7c, where it is proposed that SHB1 and CCA1/LHY cooperatively induce PIF4 expression after dawn, to regulate light, clock and temperature responses. For the reasons outlined below, I believe the data presented lacks sufficient depth to represent a significant advance in the understanding of the molecular mechanisms of light and clock control of photo/thermomorphogenesis.

(1) Key molecular and genetic studies are performed in the dark to light transition during deetiolation (Figures 1, 2, 4), in response to high temperature in continuous light (Figure 3), in yeast (Figure 5), in Nicotiana leaf cells (Figure 6), or in Mesophyll protoplasts (Figure 7). Therefore, none of the key experiments are presented in the physiological context presented in the model (Figure 7c), under diurnal/photoperiodic conditions. While the reported mechanism can be

certainly operating after dawn as authors claim, no evidence is provided, and therefore, the biological relevance of this mechanism under these conditions is absent. To establish a connection between light and the clock at dawn, at least key genetic and molecular experiments (phenotype, PIF4 expression and CHIP) must be performed in a cycling/diurnal context.

(2) To make a strong case, authors must provide additional genetic evidence that the identified molecular interaction between SHB1 and CCA1/LHY is biologically relevant. For example, authors interpret data in Figure 7b as a synergistic interaction between SHB1 and CCA1 to induce PIF4 expression. However, without any further analysis, it looks like an additive effect, which could also be interpreted as a consequence of independent activities of CCA1 and SHB1 over PIF4 expression. To assess the biological implications of the described molecular interaction, genetic evidence must be provided. For example, can the authors analyze a higher order *shb1 cca1 lhy* triple mutant for hypocotyl elongation and PIF4 expression?

(3) An important piece of evidence for the mechanism claimed by the authors is the result in Figures 4b and 4d, where authors determine that there is elimination of SHB1 binding to PIF4 promoter in *cca1 lhy* double mutant. While this is certainly a possibility, several other interpretations of the data are feasible. First, it appears that the authors used different SHB1 constructs (SHB1:YFP in 4b vs SHB1:GFP in 4d), so it is possible that these constructs determine differential binding, or that they show differential immunoprecipitation efficiencies. Even if the constructs used were the same, the expression level of the transgene in different lines might be different, thus influencing the amount of immunoprecipitated chromatin. Finally, even if the construct and the expression levels were the same in the analyzed lines, the data lack statistical analysis, and therefore, the significance of the enrichment cannot be assessed. Indeed, authors conclude that SHB1 binding is eliminated in *cca1 lhy* double mutant, but in 4d some enrichment is observed for the 4-3 genomic fragment.

(4) Because PIF4 expression is not measured in the lines analyzed in Figure 4 (SHB1:YFP and SHB1:GFP/*cca1 lhy*), it is not possible to assess the relevance of the claimed SHB1 binding to PIF4 promoter in the regulation of PIF4 expression. In addition, to establish the relevance of the model, it is required to establish molecular and phenotypic associations in these lines (i.e. what is the hypocotyl elongation phenotype?). These measurements would also contribute to validate the model. Also, how do the authors establish that SHB1:YFP and SHB1:GFP constructs produce a functional protein?

Other relatively minor points:

(5) For additional support that SHB1 does not bind directly to the PIF4 promoter, as it requires CCA1/LHY1, can the authors test whether SHB1 can bind PIF4 promoter in Y1H assays?

(6) Figure s7: Not cited in the text. Also, how do the authors interpret the several bands appearing in the input samples of SHB1:YFP/CCA1:FLAG, not present in CCA1:FLAG lines, in the western blot anti-GFP? If there is no reason stated, one single band corresponding to SHB1:YFP fusion is expected.

(7) Line 53: "TOC1 and PRRs positively regulate the expression of CCA1 and LHY, and CCA1 and LHY in turn repress the expression of TOC1 (Alabadí et al., 2001)." This sentence must be updated to the currently accepted understanding of the plant circadian clock, which places TOC1 as a general repressor of clock-gene expression (Huang et al. (2012) *Science* 336, 75–79; Gendron et al. (2012) *Proc. Natl. Acad. Sci. USA* 109, 3167–3172; Pokhilko et al. (2012) *Mol. Syst. Biol.* 8, 574).

(8) Figure s2: Authors conclude that CCA1 and LHY are required for the rhythmic expression of PIF4, and SHB1 enhances but does not alter the pattern of PIF4 rhythmic expression. However, this is only based on the gain of function transgenic line (*shb1-D*). I wonder what would happen in

the shb1 mutant.

(9) Regarding the shb1-D, it would be informative if authors discuss the type of molecular lesion of this gain of function mutant, and how this gained function is related to the molecular-phenotypic observations.

(10) Figure 3d: missing statistics.

(11) For phenotypic experiments, authors must show images of representative seedlings, so that the reader can interpret the hypocotyl length phenotype in the context of the whole seedling.

(12) Light intensity: usually expressed as $\mu\text{mol}/\text{m}^2\text{s}$ (authors use in the text $\mu\text{mol}/\text{m}^2\text{s}^2$).

(13) Figure S1: Authors claim that SHB1 only regulates PIF4, not other PIFs. However, some regulation is observed for PIF7, as PIF7 levels appear lower in shb1 mutant compared to Col wild type.

(14) Line 143: The sections in figure s2 (b,c) are switched in the text and in the legend.

(15) Figure S3 is not cited in the text.

Reviewer #1

The author claimed that this mechanism provides a negative regulation of photomorphogenesis, de-sensitization of light response, and enhancement of thermomorphogenesis. However, another way of interpretation is that transcription of PIF4 allows plants to monitor day-time shade and temperature, which alters PIF4 protein accumulation and activity post-translationally.

We added the alternative hypothesis in the first paragraph of the discussion, starting from line 407.

Fig 2a: The *shb1-D* mutation still increased hypocotyl length in the *pif4* mutant background, indicating that SHB1 can promote hypocotyl elongation independent of PIF4. Possibility of additional PIF4-independent mechanism for SHB1 action should be discussed.

Added discussion starting from line 169.

Fig 2c: Same 35S-PIF4/*cca1/lhy* lines should be crossed to wild type and *cca1/lhy*, to avoid T-DNA positional effect on PIF4 expression level.

We crossed all 3 lines from Ws to cca1 lhy.

Fig 3 shows that heat activation of PIF4 expression and thermomorphogenesis in the *shb1* and *cca1/lhy* mutant. While the hypocotyl lengths are affected in the mutants, the fold of induction by warm temperature seems unaffected, except for Fig 3a which shows same hypocotyl length between *shb1* and wild type at 22 C (but this is different from figure 2a). Should discuss clearly that SHB1 and CCA1 are not required for thermo-activation of PIF4, but the overall increase of PIF4 level increases the magnitude of heat response.

Figure 3a shows the hypocotyl length of Col and shb1 at 22 C, whereas Figure 2a shows the hypocotyl length of Ws and pif4 under red light.

The hypocotyl length was barely affected in shb1 (Supplementary Figure 2a) although the expression of PIF4 was affected in shb1 (Figure 1a and 1b). We added a paragraph starting from line 176. In shb1, a T-DNA is inserted at the 8th intron of the SHB1 gene (SALK_128406), and a truncated message was still produced (Supplementary Figure 2c and 2d). Due to the extremely low level of SHB1 expression and the low sensitivity of Taq polymerase used for semi-quantitative PCR analysis, we previously failed to detect the truncated message (Kang and Ni, 2006). Therefore, shb1 was a knock-down allele instead of a knock-out allele as we previously reported. The full-length SHB1 protein contains 745 amino acids, and shb1 lacks the C-terminal 223 amino acids but retains the N-terminal 522 amino acids. Since the N-terminus of SHB1 is important for its function (Zhou and Ni, 2010), the truncated SHB1 in shb1 is partially functional.

We added the discussion suggested by this reviewer in line 229.

Fig 4 shows CCA1/LHY-dependent SHB1 binding to PIF4 promoter in vivo. The results are clear. Why is there no binding in the dark? How does red light induce CCA1/LHY binding to PIF4 promoter? It would be nice to answer these important mechanistic questions.

The levels of CCA1:FLAG, LHY:MYC and SHB1:YFP were comparable in the dark and under red light for 3 hr when we sampled for ChIP analysis (Complementary Figure 4d). We added a paragraph in line 262 to relate this to EC activity. LUX recognizes a consensus element GATWCG and two different types of degenerate elements, GATWYG or GATWCK, in the PIF4

promoter (Nusinow et al., 2011). One consensus element, GATWCG, is 210 bp from the CCA1/LHY binding element. One degenerate element, GATTCC, is 90 bp apart from the CCA1/LHY binding element, whereas another degenerate element, GATTTG, is 1 bp apart from the forward primer of the 4-3 fragment. In ChIP analysis with ELF4:GFP, ELF4 was preferentially associated with the PIF4 promoter in the dark (Supplementary Fig. 5a). However, GBD:LUX or GBD:NOX did not interfere with the binding of GAD:CCA1 to a DNA fragment that contains both EC and CCA1 binding sites in the PIF4 promoter in a Y1H assay (Supplementary Fig. 5c, d). We speculate that the occupation of the entire PIF4 promoter by the entire EC complex and additional EC-interacting proteins in the dark may interfere with the binding of CCA1 and LHY to the PIF4 promoter.

Fig 5 shows CCA1 and LHY binding to PIF4 promoter element in Y1H assays. It would be nice to show SHB1 binding the DNA only in the presence of CCA1 in such assays (SHB1-AD + CCA1+reporter).

We have been unable to clone SHB1 cDNA in most popular vectors and SHB1 cDNA is always in a truncated form such as deletion or inversion.

Fig 6 shows BiFC and co-IP data for SHB1-CCA1/LHY interaction in plant cells. Is the interaction affected by darkness and red light?

We also detected interaction of SHB1 with either CCA1 or LHY by BiFC assays in the dark although less frequently (Supplementary Figure 7a). We performed Co-IP experiments with 35S::SHB1:GFP transgenic plants in the dark and under red light, and detected CCA1 or LHY protein by anti-CCA1 or anti-LHY antibodies (Figure S7b). Equal amount of CCA1 or LHY was precipitated by SHB1:GFP in the dark and under red light.

Fig 7: Reporter gene assays show activation of PIF4 promoter expression by SHB1, CCA1, and combination of the two. The author claimed that CCA1 and LHY act synergistically with SHB1. However, no data was shown for LHY+SHB1 and the data for CCA1+SHB1 shows additive not synergistic (more than the sum of two individually) effect. It's interesting that CCA1 activity is independent of light but that of SHB1 is, and LHY has no activity. Completing this experiment with LHY+SHB1, and testing light effect on CCA1/LHY interaction with SHB1 (Fig 6) could potentially reveal important mechanisms and answer a major outstanding question.

We removed the word "synergistically". The previous transient system was likely overloaded with effector proteins since there are endogenous CCA1, LHY and SHB1 driven by their own promoter. We delivered additional copies of each driven by the 35S promoter. In the new system, we delivered CCA1 and LHY driven by their native promoters and SHB1 driven by a 35S promoter to cca1 lhy (Figure 6b). We used 35S promoter for SHB1 since its native promoter drives a very low level of SHB1 expression and SHB1 is not rhythmically expressed (Supplementary Figure 1c). In this new system, both CCA1 and LHY has a weak transactivation activity, and SHB1 mediates red light induction which is blocked in the phyB-9 mutant (Figure 6c). LHY plays a less prominent role than that of CCA1.

Fig 7c: in the diagram, the curves for CCA1 and PIF4 don't have a 24-hr period. The enhancement of PIF4 transcription by SHB1 is not shown (e.g. by a thicker arrow).

Revised.

Language can be improved.

Professionally edited by American Journal experts.

Overall, answering any of the following questions could significantly enhance the manuscript: 1. How does red light activate SHB1? Is SHB1 accumulation, nuclear localization, or interaction with CCA1 dependent on phytochrome signaling? 2. How important is SHB1 for CCA1/LHY function beyond PIF4 regulation? How are the circadian profile of PIF4 RNA or other clock-regulated genes (or reporter gene) affected in the *shb1* and *shb1-D* mutants? 3. Is SHB1 important for thermo-adaptation and heat tolerance?

*We examined the accumulation SHB1:MYC, nuclear localization of SHB1:GFP and the interaction of SHB1:GFP with CCA1 or LHY in the dark and under red light. In addition, we designed a new transactivation assay system and performed transactivation assays in wild type and *phyB-9*. We presented the circadian profile of PIF4 RNA and protein level in the *shb1* and *shb1-D* mutants. We also explored why CCA1, LHY and SHB1 are unable to associate with the PIF4 promoter in the dark.*

Reviewer #2

1. Statistical assessments should be applied to all figures involving qRT-PCR.

Added.

2. In Fig. 2a, *shb1 pif4* double mutant showed intermediate phenotype between *shb1* and *pif4* single mutants. This indicated that the actions of SHB1 do not entirely go through PIF4? Therefore, the conclusion that PIF4 functioning downstream of SHB1 needs to be either toned down or alternative explanations should be provided/discussed.

We changed the section title as genetic interaction of PIF4 with SHB1 and CCA1. Alternative explanation is provided in line 169.

3. Whether PIF4 functions downstream of CCA1/LHY? In Fig. 2c, the overexpression of PIF4 in *cca1 lhy* double mutants resulted in elongated phenotype. The authors concluded that PIF4 acts downstream of CCA1 and LHY. In fact, based on this data, PIF4 could also function in an independent pathway to promote hypocotyl elongation. It might be helpful to combine the results in Fig. 1 (the R-induced PIF4 relies on CCA1/LHY) in concluding that PIF4 acts downstream of CCA1/LHY.

Revised as suggested in line 195.

4. As shown in Fig. 3b and 3d, hypocotyls of *cca1 lhy* double mutant still elongate in response to high temperature and PIF4 was still upregulated under high temperature in *cca1 lhy*. If the data were shown in fold changes, perhaps the degree of up-regulation was comparable between wild-type and the mutants. This indicated that, under high temperature, the induction of PIF4 does not entirely rely on CCA1/LHY? Similarly results for PIF4 up-regulation and elongated hypocotyl were also observed in *shb1* mutant (Fig. 3a, 3d), indicating the induction of PIF4 under higher temperature is not entirely dependent on SHB1?

Revised as suggested in line 211 and line 218.

5. To claim SHB1 binds to the promoter of PIF4, SHB1-YFP was used. It was, however, unclear whether SHB1:YFP was biologically functional? Was functional complementation performed? Were overexpression lines used? What was the ZT used for ChIP? Also, ChIP should be presented as % input not normalized to UBQ10. For a meaningful comparison in Fig. 4d, the expression level of SHB1 in WT and *cca1 lhy* mutant should be carefully quantitated and used for normalization?

*For consistence, 35S::SHB1:GFP line was used for ChIP analysis. This construct in *shb1* caused a longer hypocotyl (Supplementary Figure 4a). The samples were collected at ZT3 (8 am as ZT0) for dark-grown plants without and with 3 hr red light treatment. In regard to ChIP normalization, we discuss whether we might stay with the current way for the following reasons. First, we found comparable level of SHB1:GFP in *Ws* and *cca1 lhy* (Supplementary Figure 4d). Second, different manipulations are involved for input or ChIP samples such as harvest efficiency by antibody, wash, elution, cross-link reversal and DNA precipitation. The normalization to UBQ10 ensures a consideration of different efficiencies in genomic DNA recovery. Third, ChIP-seq normally uses input for normalization since the mock ChIP does not recover enough DNA for sequencing. Last, additional normalization to the level of SHB1 protein biases against the mock ChIP sample since it is difficult to normalize control samples in the absence of antibodies. Under these circumstances, the ChIP data are not connected to the level of SHB1:GFP protein.*

6. The protein-protein interaction data shown in Fig. S6: for those showing no protein-protein interactions, did the constructs yield protein products with comparable expression levels? Also, as shown in Fig. 6 and S6, LHY could interact with SHB1. However, LHY cannot transactivate PIF4 promoter (Fig. 7). Whether the LHY-SHB1 interaction possesses biological significance should at least be discussed.

We transformed tobacco with the various constructs as shown in Supplementary Figures 8 and 9, isolated total proteins, and probed YFP-N fusion proteins with anti-MYC and YFP-C fusion proteins with anti-HA antibodies. All truncated proteins were expressed at comparable levels (Supplementary Figure 10a and 10b). We designed a new transactivation assay system and defined a less prominent role for LHY (see our response to comments on Figure 7 by reviewer 1).

7. For the transactivation assays shown in Fig. 7b, the transfection of the effector 35S::SHB1 showed R-dependent transactivation activities whereas 35S::CCA1 has comparable transactivation activity toward PIF4::LUC reporter construct under dark and R? Were the SHB1 protein levels different between protoplasts under dark and red light conditions? How about the CCA1 protein levels? The Myc tag for the effector proteins should allow the detection of the effector proteins in the protoplasts. The transactivation activities of different effector proteins could also be normalized by the levels of the effector proteins. For example, is CCA1 indeed much more effective in triggering PIF4 expression, or it is simply CCA1 protein accumulates to a higher level than LHY?

The SHB1 protein levels are not different between protoplasts in the dark and under red light (Supplementary Figure 10b). The level of CCA1 protein is also comparable with that of LHY. We discuss here whether we should normalize the transactivation activities of different effector proteins by the levels of the effector proteins. First, there is no effective way to normalize the control where only the reporter delivered but not any of the effector proteins. Second, there are

three different effector proteins of different sizes. Normalization of the reporter LUC activity to the reference REN activity bearing on the same plasmid might still be an option.

8. A time course binding of CCA1/SHB1 to PIF4 promoter could be examined to better support the model shown in Fig. 7c? In addition, PIF4 protein should accumulate during the day through the action of CCA1-SHB1 complex in triggering PIF4 expression for thermomorphogenic responses. Previous reports showed overexpressed PIF4 was effectively degraded after short period of R light treatment (Kumar et al 2012). On the other hand, Zhang et al 2017 and Lorrain et al 2008 showed that PIF4 re-accumulates after a few hours of light treatment, in line better with the model shown in Fig. 7c. Is CCA1-SHB1 complex responsible for the reappearance of PIF4 protein in the later cases?

We performed rhythmic ChIP assay with 35S::SHB1:GFP transgenic plants grown under 12 hr red light and 12 hr dark, and started sampling at ZT0 and every 3 hr thereafter for 24 hrs (Figure 4d). We used 35S::SHB1:GFP plants since SHB1 native promoter is very weak and SHB1 expression is not rhythmic. ChIP was performed by using anti-CCA1, anti-LHY and anti-GFP antibodies. We examined rhythmic PIF4 protein accumulation in Col. Shb1, Ws and shb1-D in Figure 1d. PIF4 protein was monitored by using anti-PIF4 antibody. The seedlings were first grown under 12 hr dark and 12 hr 2 $\mu\text{mol}/\text{m}^2/\text{s}$ red light for 5 days. We started sampling at ZT0 and every 3 hr thereafter for 24 hrs. To better correlate SHB1 ChIP and PIF4 expression, we only presented rhythmic PIF4 expression in Figure 4e.

Minor points:

1. Fig. S2b and S2c is reversed in order?

Revised.

2. Line 53-55, TOC1 has been shown to be direct repressors of CCA1. The description should be corrected.

Corrected.

3. In Fig. S1, the red light induced upregulation of PIF7 was also compromised in shb1 mutant. The description in line 120-122 should be slightly modified to reflect the result?

Revised.

4. Line 158, Fig 2b should be cited.

Cited.

5. Line 206, despite marked in manuscript text, no data for M2 could be found in Fig. 5c.

Data for M2 added in Figure S6.

6. The band corresponded to SHB1-GFP should be marked in Fig. 6c.

We performed new co-IP experiments showing a relatively strong SHB1:GFP band. We did mark the SHB1:GFP band in Figure S11 where we showed the full image.

7. Line 234, Fig. S5a should be cited instead of S6a.

Revised.

8. Line 259, “CCA1 or LHY likely possesses a partial trans-activation activity beside its ability to bind DNA”. It was unclear what this statement was referring to?

This sentence was deleted.

Reviewer #3

(1) To establish a connection between light and the clock at dawn, at least key genetic and molecular experiments (phenotype, PIF4 expression and ChIP) must be performed in a cycling/diurnal context.

We presented rhythmic ChIP and PIF4 expression in Figure 4d and 4e (see response to comments 7 and 8 by reviewer 2). We examined PIF4 expression and accumulation in Col, shb1, Ws and shb1-D seedlings (Figure 1b to 1d). We don't have the device and capacity to follow the cycling and diurnal hypocotyl elongation response.

(2) To assess the biological implications of the described molecular interaction, genetic evidence must be provided. For example, can the authors analyze a higher order shb1 cca1 lhy triple mutant for hypocotyl elongation and PIF4 expression?

A triple mutant was constructed and analyzed for hypocotyl elongation and PIF4 expression in Col, shb1, Ws, cca1 lhy, Col/Ws, shb1 cca1 lhy (Supplementary Figure 2a and 2b). However, shb1 is a weak and knock-down allele (see response to comments on Figure 3 by reviewer 1) and not much severe phenotype was observed in the triple mutant.

(3) There is elimination of SHB1 binding to PIF4 promoter in cca1 lhy double mutant. While this is certainly a possibility, several other interpretations of the data are feasible. First, it appears that the authors used different SHB1 constructs (SHB1:YFP in 4b vs SHB1:GFP in 4d), so it is possible that these constructs determine differential binding, or that they show differential immunoprecipitation efficiencies. Even if the constructs used were the same, the expression level of the transgene in different lines might be different, thus influencing the amount of immunoprecipitated chromatin. Finally, even if the construct and the expression levels were the same in the analyzed lines, the data lack statistical analysis, and therefore, the significance of the enrichment cannot be assessed. Indeed, authors conclude that SHB1 binding is eliminated in cca1 lhy double mutant, but in 4d some enrichment is observed for the 4-3 genomic fragment.

We used SHB1:GFP transgene in Ws and cca1 lhy background and added statistical analysis (Figure 4b and 4c). The enrichment for the 4-3 genomic fragment in cca1 lhy is not statistically significant. We also examined SHB1:GFP protein accumulation in Ws and cca1 lhy (Supplementary Figure 4d).

(4) Because PIF4 expression is not measured in the lines analyzed in Figure 4 (SHB1:YFP and SHB1:GFP/cca1 lhy), it is not possible to assess the relevance of the claimed SHB1 binding to PIF4 promoter in the regulation of PIF4 expression. In addition, to establish the relevance of the model, it is required to establish molecular and phenotypic associations in these lines (i.e. what is the hypocotyl elongation phenotype?). These measurements would also contribute to validate the model. Also, how do the authors establish that SHB1:YFP and SHB1:GFP constructs produce a functional protein?

We presented the hypocotyl elongation phenotype and PIF4 expression for the 35S::SHB1:GFP transgenic plants in Ws and cca1 lhy in the dark and under red light at the time points when we

sampled for ChIP (Supplementary Figure 4e and 4f). We also reported the hypocotyl phenotype and PIF4 expression in the 35S::SHB1:GFP lines in shb1 (Supplementary Figure 4a and 4b).

Other relatively minor points:

(5) For additional support that SHB1 does not bind directly to the PIF4 promoter, as it requires CCA1/LHY1, can the authors test whether SHB1 can bind PIF4 promoter in Y1H assays?
We have been unable to clone SHB1 cDNA to most popular vectors.

(6) Figure s7: Not cited in the text. Also, how do the authors interpret the several bands appearing in the input samples of SHB1:YFP/CCA1:FLAG, not present in CCA1:FLAG lines, in the western blot anti-GFP? If there is no reason stated, one single band corresponding to SHB1:YFP fusion is expected.

Cited. Figure S11 replaced the original Figure S7. The additional bands in SHB1:GFP lane may represent degradation products of SHB1:GFP.

(7) Line 53: “TOC1 and PRRs positively regulate the expression of CCA1 and LHY, and CCA1 and LHY in turn repress the expression of TOC1 (Alabadi et al., 2001).” This sentence must be updated to the currently accepted understanding of the plant circadian clock, which places TOC1 as a general repressor of clock-gene expression (Huang et al. (2012) Science 336, 75–79; Gendron et al. (2012) Proc. Natl. Acad. Sci. USA 109, 3167–3172; Pokhilko et al. (2012) Mol. Syst. Biol. 8, 574).

We updated to the current model.

(8) Figure s2: Authors conclude that CCA1 and LHY are required for the rhythmic expression of PIF4, and SHB1 enhances but does not alter the pattern of PIF4 rhythmic expression. However, this is only based on the gain of function transgenic line (shb1-D). I wonder what would happen in the shb1 mutant.

We added PIF4 rhythmic expression in Col and shb1 under red light (Figure 1b).

(9) Regarding the shb1-D, it would be informative if authors discuss the type of molecular lesion of this gain of function mutant, and how this gained function is related to the molecular-phenotypic observations.

We introduced shb1-D in line 100.

(10) Figure 3d: missing statistics.

Added statistical analysis.

(11) For phenotypic experiments, authors must show images of representative seedlings, so that the reader can interpret the hypocotyl length phenotype in the context of the whole seedling.

We added photos for the hypocotyl phenotypes.

(12) Light intensity: usually expressed as $\mu\text{mol}/\text{m}^2/\text{s}$ (authors use in the text $\mu\text{mol}/\text{m}^2\text{s}^2$).

Corrected.

(13) Figure S1: Authors claim that SHB1 only regulates PIF4, not other PIFs. However, some regulation is observed for PIF7, as PIF7 levels appear lower in shb1 mutant compared to Col wild type.

Revised in line 127.

(14) Line 143: The sections in figure s2 (b,c) are switched in the text and in the legend.

Corrected.

(15) Figure S3 is not cited in the text.

Cited.

Reviewer #1

The author claimed that this mechanism provides a negative regulation of photomorphogenesis, de-sensitization of light response, and enhancement of thermomorphogenesis. However, another way of interpretation is that transcription of PIF4 allows plants to monitor day-time shade and temperature, which alters PIF4 protein accumulation and activity post-translationally.

We added the alternative hypothesis in the first paragraph of the discussion, starting from line 441.

Fig 2a: The *shb1-D* mutation still increased hypocotyl length in the *pif4* mutant background, indicating that SHB1 can promote hypocotyl elongation independent of PIF4. Possibility of additional PIF4-independent mechanism for SHB1 action should be discussed.

Added discussion starting from line 170.

Fig 2c: Same 35S-PIF4/*cca1/lhy* lines should be crossed to wild type and *cca1/lhy*, to avoid T-DNA positional effect on PIF4 expression level.

We crossed all 3 lines from Ws to cca1 lhy.

Fig 3 shows that heat activation of PIF4 expression and thermomorphogenesis in the *shb1* and *cca1/lhy* mutant. While the hypocotyl lengths are affected in the mutants, the fold of induction by warm temperature seems unaffected, except for Fig 3a which shows same hypocotyl length between *shb1* and wild type at 22 C (but this is different from figure 2a). Should discuss clearly that SHB1 and CCA1 are not required for thermo-activation of PIF4, but the overall increase of PIF4 level increases the magnitude of heat response.

Figure 3a shows the hypocotyl length of Col and shb1 at 22 C, whereas Figure 2a shows the hypocotyl length of Ws and pif4 under red light.

The hypocotyl length was barely affected in shb1 (Supplementary Figure 2a) although the expression of PIF4 was affected in shb1 (Figure 1a and 1b). We added a paragraph starting from line 190. In shb1, a T-DNA is inserted at the 8th intron of the SHB1 gene (SALK_128406), and a truncated message was still produced (Supplementary Fig. 2d, e). Given the extremely low level of SHB1 expression and the low sensitivity of Taq polymerase used for semi-quantitative PCR analysis, we previously failed to detect the truncated message (Kang and Ni, 2006).

Therefore, shb1 was a partial loss-of-function allele instead of a knockout allele. The full-length SHB1 protein contains 745 amino acids, and shb1 lacks the C-terminal 223 amino acids but retains the N-terminal 522 amino acids. Given that the N-terminus of SHB1 is important for its function (Zhou and Ni, 2010), the truncated SHB1 in shb1 is partially functional.

We have been unable to identify a true shb1 knockout over many years, and shb1 knockout may cause a lethal phenotype.

We added the discussion suggested by this reviewer in line 228.

Fig 4 shows CCA1/LHY-dependent SHB1 binding to PIF4 promoter in vivo. The results are clear. Why is there no binding in the dark? How does red light induce CCA1/LHY binding to PIF4 promoter? It would be nice to answer these important mechanistic questions.

The levels of CCA1:FLAG, LHY:MYC and SHB1:YFP were comparable in the dark and under red light for 3 hr when we sampled for ChIP analysis (Complementary Figure 4d). We added a paragraph in line 262 to relate this to EC activity. LUX recognizes a consensus element

GATWCG and two different types of degenerate elements, GATWYG or GATWCK, in the PIF4 promoter (Nusinow et al., 2011). One consensus element, GATWCG, is 210 bp from the CCA1/LHY binding element. One degenerate element, GATTCC, is 90 bp apart from the CCA1/LHY binding element, whereas another degenerate element, GATTTG, is 1 bp apart from the forward primer of the 4-3 fragment. In ChIP analysis with ELF4:GFP, ELF4 was preferentially associated with the PIF4 promoter in the dark (Supplementary Fig. 5a). However, GBD:LUX or GBD:NOX did not interfere with the binding of GAD:CCA1 to a DNA fragment that contains both EC and CCA1 binding sites in the PIF4 promoter in a Y1H assay (Supplementary Fig. 5c, d). We speculate that the occupation of the PIF4 promoter by the entire EC complex and additional EC-interacting proteins in the dark may interfere with the binding of CCA1 and LHY to the PIF4 promoter.

Fig 5 shows CCA1 and LHY binding to PIF4 promoter element in Y1H assays. It would be nice to show SHB1 binding the DNA only in the presence of CCA1 in such assays (SHB1-AD + CCA1+reporter).

We have been unable to clone SHB1 cDNA in most popular vectors over many years. SHB1 cDNA is always in a truncated form such as deletion or inversion.

Fig 6 shows BiFC and co-IP data for SHB1-CCA1/LHY interaction in plant cells. Is the interaction affected by darkness and red light?

We also detected interaction of SHB1 with either CCA1 or LHY by BiFC assays in the dark although less frequently (new Supplementary Figure 9a). We performed Co-IP experiments with 35S::SHB1:GFP transgenic plants in the dark and under red light, and detected CCA1 or LHY protein by anti-CCA1 or anti-LHY antibodies (new Supplementary Figure 9b). Equal amount of CCA1 or LHY was precipitated by SHB1:GFP in the dark and under red light.

Fig 7: Reporter gene assays show activation of PIF4 promoter expression by SHB1, CCA1, and combination of the two. The author claimed that CCA1 and LHY act synergistically with SHB1. However, no data was shown for LHY+SHB1 and the data for CCA1+SHB1 shows additive not synergistic (more than the sum of two individually) effect. It's interesting that CCA1 activity is independent of light but that of SHB1 is, and LHY has no activity. Completing this experiment with LHY+SHB1, and testing light effect on CCA1/LHY interaction with SHB1 (Fig 6) could potentially reveal important mechanisms and answer a major outstanding question.

We removed the word "synergistically". The previous transient system was likely overloaded with effector proteins since there are endogenous CCA1, LHY and SHB1 driven by their own promoter. We delivered additional copies of each driven by the 35S promoter. In the new system, we delivered CCA1 and LHY driven by their native promoters and SHB1 driven by a 35S promoter to cca1 lhy (Figure 6e). We used 35S promoter for SHB1 since its native promoter drives a very low level of SHB1 expression and SHB1 is not rhythmically expressed (Supplementary Figure 1c). In this new system, both CCA1 and LHY has a weak transactivation activity, and SHB1 mediates red light induction which is blocked in the phyB-9 mutant (Figure 6f). LHY plays a less prominent role than that of CCA1.

Fig 7c: in the diagram, the curves for CCA1 and PIF4 don't have a 24-hr period. The enhancement of PIF4 transcription by SHB1 is not shown (e.g. by a thicker arrow).

Revised.

Language can be improved.

Professionally edited by American Journal experts.

Overall, answering any of the following questions could significantly enhance the manuscript: 1. How does red light activate SHB1? Is SHB1 accumulation, nuclear localization, or interaction with CCA1 dependent on phytochrome signaling? 2. How important is SHB1 for CCA1/LHY function beyond PIF4 regulation? How are the circadian profile of PIF4 RNA or other clock-regulated genes (or reporter gene) affected in the *shb1* and *shb1-D* mutants? 3. Is SHB1 important for thermo-adaptation and heat tolerance?

1) We examined the accumulation SHB1:MYC, nuclear localization of SHB1:GFP and the interaction of SHB1:GFP with CCA1 or LHY in the dark and under red light. In addition, we designed a new transactivation assay system and performed transactivation assays in wild type and phyB-9. 2) We presented the circadian profile of PIF4 RNA and protein level in the shb1 and shb1-D mutants. We also explored why CCA1, LHY and SHB1 are unable to associate with the PIF4 promoter in the dark. 3) We introduced pPIF4::PIF4 with a wild type, base-substituted or deleted M1 box to pif4 or shb1-D pif4. Their light- and thermo-responses, PIF4 expression and in vivo ChIP analysis were presented in Figure 5, Figure S7 and Figure S8. Regulated PIF4 expression by SHB1-CCA1 only contributes partially to PIF4-mediated thermo-responses.

Reviewer #2

1. Statistical assessments should be applied to all figures involving qRT-PCR.

Added.

2. In Fig. 2a, *shb1 pif4* double mutant showed intermediate phenotype between *shb1* and *pif4* single mutants. This indicated that the actions of SHB1 do not entirely go through PIF4? Therefore, the conclusion that PIF4 functioning downstream of SHB1 needs to be either toned down or alternative explanations should be provided/discussed.

We changed the section title as genetic interaction of PIF4 with SHB1 and CCA1. Alternative explanation is provided in line 170.

3. Whether PIF4 functions downstream of CCA1/LHY? In Fig. 2c, the overexpression of PIF4 in *cca1 lhy* double mutants resulted in elongated phenotype. The authors concluded that PIF4 acts downstream of CCA1 and LHY. In fact, based on this data, PIF4 could also function in an independent pathway to promote hypocotyl elongation. It might be helpful to combine the results in Fig. 1 (the R-induced PIF4 relies on CCA1/LHY) in concluding that PIF4 acts downstream of CCA1/LHY.

Revised as suggested in line 183.

4. As shown in Fig. 3b and 3d, hypocotyls of *cca1 lhy* double mutant still elongate in response to high temperature and PIF4 was still upregulated under high temperature in *cca1 lhy*. If the data were shown in fold changes, perhaps the degree of up-regulation was comparable between wild-type and the mutants. This indicated that, under high temperature, the induction of PIF4 does not entirely rely on CCA1/LHY? Similarly results for PIF4 up-regulation and elongated hypocotyl

were also observed in *shb1* mutant (Fig. 3a, 3d), indicating the induction of PIF4 under higher temperature is not entirely dependent on SHB1?

Revised as suggested in line 213 and line 228.

5. To claim SHB1 binds to the promoter of PIF4, SHB1-YFP was used. It was, however, unclear whether SHB1:YFP was biologically functional? Was functional complementation performed? Were overexpression lines used? What was the ZT used for ChIP? Also, ChIP should be presented as % input not normalized to UBQ10. For a meaningful comparison in Fig. 4d, the expression level of SHB1 in WT and *cca1 lhy* mutant should be carefully quantitated and used for normalization?

*For consistence, 35S::SHB1:GFP line was used for ChIP analysis. This construct in *shb1* caused a longer hypocotyl (Supplementary Figure 4a). The samples were collected at ZT3 (8 am as ZT0) for dark-grown plants without and with 3 hr red light treatment. In regard to ChIP normalization, we discuss whether we might stay with the current way for the following reasons. First, we found comparable level of SHB1:GFP in *Ws* and *cca1 lhy* (Supplementary Figure 4d). Second, different manipulations are involved for input or ChIP samples such as harvest efficiency by antibody, wash, elution, cross-link reversal and DNA precipitation. The normalization to UBQ10 ensures a consideration of different efficiencies in genomic DNA recovery. Third, ChIP-seq normally uses input for normalization since the mock ChIP does not recover enough DNA for sequencing. Last, additional normalization to the level of SHB1 protein biases against the mock ChIP sample since it is difficult to normalize control samples in the absence of antibodies. Under these circumstances, the ChIP data are not connected to the level of SHB1:GFP protein.*

6. The protein-protein interaction data shown in Fig. S6: for those showing no protein-protein interactions, did the constructs yield protein products with comparable expression levels? Also, as shown in Fig. 6 and S6, LHY could interact with SHB1. However, LHY cannot transactivate PIF4 promoter (Fig. 7). Whether the LHY-SHB1 interaction possesses biological significance should at least be discussed.

We transformed tobacco with the various constructs as shown in Supplementary Figures 10 and 11, isolated total proteins, and probed YFP-N fusion proteins with anti-MYC and YFP-C fusion proteins with anti-HA antibodies. All truncated proteins were expressed at comparable levels (Supplementary Figure 12a and 12b). We designed a new transactivation assay system and defined a less prominent role for LHY (see our response to comments on Figure 7 by reviewer 1).

7. For the transactivation assays shown in Fig. 7b, the transfection of the effector 35S::SHB1 showed R-dependent transactivation activities whereas 35S::CCA1 has comparable transactivation activity toward PIF4::LUC reporter construct under dark and R? Were the SHB1 protein levels different between protoplasts under dark and red light conditions? How about the CCA1 protein levels? The Myc tag for the effector proteins should allow the detection of the effector proteins in the protoplasts. The transactivation activities of different effector proteins could also be normalized by the levels of the effector proteins. For example, is CCA1 indeed much more effective in triggering PIF4 expression, or it is simply CCA1 protein accumulates to a higher level than LHY?

The SHB1 protein levels are not different between protoplasts in the dark and under red light (Supplementary Figure 12b). The level of CCA1 protein is also comparable with that of LHY. We

discuss here whether we should normalize the transactivation activities of different effector proteins by the levels of the effector proteins. First, there is no effective way to normalize the control where only the reporter delivered but not any of the effector proteins. Second, there are three different effector proteins of different sizes. Normalization of the reporter LUC activity to the reference REN activity bearing on the same plasmid might still be an option.

8. A time course binding of CCA1/SHB1 to PIF4 promoter could be examined to better support the model shown in Fig. 7c? In addition, PIF4 protein should accumulate during the day through the action of CCA1-SHB1 complex in triggering PIF4 expression for thermomorphogenic responses. Previous reports showed overexpressed PIF4 was effectively degraded after short period of R light treatment (Kumar et al 2012). On the other hand, Zhang et al 2017 and Lorrain et al 2008 showed that PIF4 re-accumulates after a few hours of light treatment, in line better with the model shown in Fig. 7c. Is CCA1-SHB1 complex responsible for the reappearance of PIF4 protein in the later cases?

We performed rhythmic ChIP assay with 35S::SHB1:GFP transgenic plants grown under 12 hr red light and 12 hr dark, and started sampling at ZT0 and every 3 hr thereafter for 24 hrs (Figure 4d). We used 35S::SHB1:GFP plants since SHB1 native promoter is very weak and SHB1 expression is not rhythmic. ChIP was performed by using anti-CCA1, anti-LHY and anti-GFP antibodies. We examined rhythmic PIF4 protein accumulation in Col, shb1, Ws and shb1-D in Figure 1d. PIF4 protein was monitored by using anti-PIF4 antibody. The seedlings were first grown under 12 hr dark and 12 hr 2 $\mu\text{mol}/\text{m}^2/\text{s}$ red light for 5 days. We started sampling at ZT0 and every 3 hr thereafter for 24 hrs. To better correlate SHB1 ChIP and PIF4 expression, we presented rhythmic PIF4 expression in Figure 4e.

Minor points:

1. Fig. S2b and S2c is reversed in order?

Revised.

2. Line 53-55, TOC1 has been shown to be direct repressors of CCA1. The description should be corrected.

Corrected.

3. In Fig. S1, the red light induced upregulation of PIF7 was also compromised in shb1 mutant. The description in line 120-122 should be slightly modified to reflect the result?

Revised.

4. Line 158, Fig 2b should be cited.

Cited.

5. Line 206, despite marked in manuscript text, no data for M2 could be found in Fig. 5c.

Data for M2 added in Figure S6.

6. The band corresponded to SHB1-GFP should be marked in Fig. 6c.

We performed new co-IP experiments showing a relatively strong SHB1:GFP band. We did mark the SHB1:GFP band in Figure S13 where we showed the full image.

7. Line 234, Fig. S5a should be cited instead of S6a.

Revised.

8. Line 259, “CCA1 or LHY likely possesses a partial trans-activation activity beside its ability to bind DNA”. It was unclear what this statement was referring to?

This sentence was deleted.

Reviewer #3

(1) To establish a connection between light and the clock at dawn, at least key genetic and molecular experiments (phenotype, PIF4 expression and ChIP) must be performed in a cycling/diurnal context.

We presented rhythmic ChIP and PIF4 expression in Figure 4d and 4e (see response to comments 7 and 8 by reviewer 2). We examined PIF4 expression and accumulation in Col, shb1, Ws and shb1-D seedlings (Figure 1b to 1d). We don't have the device and capacity to take the photos of dark-grown seedlings and follow the cycling and diurnal hypocotyl elongation response.

(2) To assess the biological implications of the described molecular interaction, genetic evidence must be provided. For example, can the authors analyze a higher order *shb1 cca1 lhy* triple mutant for hypocotyl elongation and PIF4 expression?

A triple mutant was constructed and analyzed for hypocotyl elongation and PIF4 expression in Col, shb1, Ws, cca1 lhy, Col/Ws, shb1 cca1 lhy (Supplementary Figure 2b and 2c). However, shb1 is a weak and knock-down allele (see response to comments on Figure 3 by reviewer 1) and not much severe phenotype was observed in the triple mutant. We generated a shb1-D cca1 lhy triple mutant. Their light responses and PIF4 expression were presented in Figure 2d, e. In contrast, a shb1-D cca1 lhy triple mutant showed a hypocotyl phenotype and PIF4 expression similar to that of the cca1 lhy double mutant (Fig. 2d, e and Supplementary Fig. 2f). Thus, the promotion of hypocotyl and PIF4 expression by SHB1 under red light critically relies on CCA1 and LHY. The thermo responses and PIF4 expression of the shb1-D cca1 lhy triple mutant were presented in Figure 3d-f. Both hypocotyl and PIF4 expression at 29°C in shb1-D cca1 lhy triple mutant were higher than that of cca1 lhy double mutant, but lower than that of shb1-D (Fig. 3d-f). In addition to CCA1/LHY, other transcription factors may participate in SHB1-mediated thermos-responses.

(3) There is elimination of SHB1 binding to PIF4 promoter in *cca1 lhy* double mutant. While this is certainly a possibility, several other interpretations of the data are feasible. First, it appears that the authors used different SHB1 constructs (SHB1:YFP in 4b vs SHB1:GFP in 4d), so it is possible that these constructs determine differential binding, or that they show differential immunoprecipitation efficiencies. Even if the constructs used were the same, the expression level of the transgene in different lines might be different, thus influencing the amount of immunoprecipitated chromatin. Finally, even if the construct and the expression levels were the same in the analyzed lines, the data lack statistical analysis, and therefore, the significance of the enrichment cannot be assessed. Indeed, authors conclude that SHB1 binding is eliminated in *cca1 lhy* double mutant, but in 4d some enrichment is observed for the 4-3 genomic fragment.

We used SHB1:GFP transgene in Ws and cca1 lhy background and added statistical analysis (Figure 4b and 4c). The enrichment for the 4-3 genomic fragment in cca1 lhy is not statistically significant. We also examined SHB1:GFP protein accumulation in Ws and cca1 lhy (Supplementary Figure 4d).

(4) Because PIF4 expression is not measured in the lines analyzed in Figure 4 (SHB1:YFP and SHB1:GFP/cca1 lhy), it is not possible to assess the relevance of the claimed SHB1 binding to PIF4 promoter in the regulation of PIF4 expression. In addition, to establish the relevance of the model, it is required to establish molecular and phenotypic associations in these lines (i.e. what is the hypocotyl elongation phenotype?). These measurements would also contribute to validate the model. Also, how do the authors establish that SHB1:YFP and SHB1:GFP constructs produce a functional protein?

We presented the hypocotyl elongation phenotype and PIF4 expression for the 35S::SHB1:GFP transgenic plants in Ws and cca1 lhy in the dark and under red light at the time points when we sampled for ChIP (Supplementary Figure 4e and 4f). We also reported the hypocotyl phenotype and PIF4 expression in the 35S::SHB1:GFP lines in shb1 (Supplementary Figure 4a and 4b).

Other relatively minor points:

(5) For additional support that SHB1 does not bind directly to the PIF4 promoter, as it requires CCA1/LHY1, can the authors test whether SHB1 can bind PIF4 promoter in Y1H assays?

We have been unable to clone SHB1 cDNA to most popular vectors over many years. The cDNA always has deletions or conversions.

(6) Figure s7: Not cited in the text. Also, how do the authors interpret the several bands appearing in the input samples of SHB1:YFP/CCA1:FLAG, not present in CCA1:FLAG lines, in the western blot anti-GFP? If there is no reason stated, one single band corresponding to SHB1:YFP fusion is expected.

Cited. Figure S13 replaced the original Figure S7. The additional bands in SHB1:GFP lane may represent degradation products of SHB1:GFP.

(7) Line 53: “TOC1 and PRRs positively regulate the expression of CCA1 and LHY, and CCA1 and LHY in turn repress the expression of TOC1 (Alabadí et al., 2001).” This sentence must be updated to the currently accepted understanding of the plant circadian clock, which places TOC1 as a general repressor of clock-gene expression (Huang et al. (2012) Science 336, 75–79; Gendron et al. (2012) Proc. Natl. Acad. Sci. USA 109, 3167–3172; Pokhilko et al. (2012) Mol. Syst. Biol. 8, 574).

We updated to the current model.

(8) Figure s2: Authors conclude that CCA1 and LHY are required for the rhythmic expression of PIF4, and SHB1 enhances but does not alter the pattern of PIF4 rhythmic expression. However, this is only based on the gain of function transgenic line (shb1-D). I wonder what would happen in the shb1 mutant.

We added PIF4 rhythmic expression in Col and shb1 under red light (Figure 1b).

(9) Regarding the *shb1-D*, it would be informative if authors discuss the type of molecular lesion of this gain of function mutant, and how this gained function is related to the molecular-phenotypic observations.

We introduced shb1-D in line 100.

(10) Figure 3d: missing statistics.

Added statistical analysis.

(11) For phenotypic experiments, authors must show images of representative seedlings, so that the reader can interpret the hypocotyl length phenotype in the context of the whole seedling.

We added photos for the hypocotyl phenotypes.

(12) Light intensity: usually expressed as $\mu\text{mol}/\text{m}^2/\text{s}$ (authors use in the text $\mu\text{mol}/\text{m}^2\text{s}^2$).

Corrected.

(13) Figure S1: Authors claim that SHB1 only regulates PIF4, not other PIFs. However, some regulation is observed for PIF7, as PIF7 levels appear lower in *shb1* mutant compared to Col wild type.

Revised in line 127.

(14) Line 143: The sections in figure s2 (b,c) are switched in the text and in the legend.

Corrected.

(15) Figure S3 is not cited in the text.

Cited.

Reviewers' comments:

Reviewer #1 (Remarks to the Author):

The revision has significantly improved the manuscript.

Line 48-56: The two back-to-back papers in Cell 1998 describing the original cloning of the first plant clock genes, CCA1 and LHY, deserve citation in this manuscript.

Line 449: "In general, CCA1 and LHY repress the evening element-containing genes at dawn," The original work discovered CCA1 as an activator of a dawn gene LHCB, This should be mentioned.

Reviewer #3 (Remarks to the Author):

In this work, Sun and co-authors identify a novel layer of regulation of PIF4 expression by SHB1 and the circadian clock components CCA1/LHY. Authors show that both CCA1 and SHB1 bind to the PIF4 promoter to induce its expression in response to red light during deetiolation and under diurnal/cycling conditions, and in response to high temperature.

Authors have extensively revised the previous version, and have added substantial new genetic and molecular evidence in support of the central conclusions. The new ChIP and gene expression data in Figures 4-6, and accompanying supplemental figures, provide a strong support that SHB1 and CCA1/LHY cooperatively bind to the promoter and induce the expression of PIF4 in response to red light and during the morning under cycling conditions.

Still, some things can be improved:

(1) Authors acknowledged that PIF4 expression is regulated by the evening complex, yet the well documented direct repression of PIF4 expression by PRR components of the circadian clock is not mentioned. Appropriate references must be included.

(2) The molecular lesion of the shb1 mutant allele must be clarified. On the basis of new qRT-PCR data (Supplemental Figure 2e) authors conclude that "shb1 was a partial loss-of-function allele instead of a knockout allele.", as they observe that "In shb1, a T-DNA is inserted at the 8th intron of the SHB1 gene (SALK_128406), and a truncated message was still produced (Supplementary Fig. 2d, e)."

However, no evidence for the presence of such a "truncated message" is provided. Authors determine by RT-PCR the presence of the 5' region of the transcript upstream of the T-DNA insertion, but this is no proof of a truncated transcript. Moreover, as the T-DNA insertion is located in an intron, there is the formal possibility that the intron is spliced correctly resulting in functional mature transcript. This point is particularly relevant as most of the functional evidence regarding SHB1 relies either in a gain of function mutant (shb1-D) or in SHB1 overexpression lines. The shb1 mutant allele certainly shows consistent reduced levels of PIF4 transcript, but no significant phenotypic growth consequences are observed in the present study. Is there evidence of shb1 mutant allele being a partial loss-of-function allele in other reported phenotypic responses?

(3) Figures 1-3: The headlines of figure legends emphasize the role of SHB1 and CCA1 on different responses. However, genetic data are always performed with cca1 lhy double mutant, and therefore, the relative role of CCA1 and LHY is not assessed. Therefore, the headlines may not arbitrarily focus on CCA1, and must also include LHY (e.g. CCA1/LHY).

(4) Moreover, in Discussion (line 433): "Our study suggests that the highly expressed circadian clock protein CCA1 in the morning recruits SHB1 to activate red light-induced PIF4 expression and desensitize light responses. LHY may play a less prominent role in this process."

I don't think there is strong evidence in the article that suggests that LHY has a minor role compared to CCA1. In order to make this claim, it would need a closer look. The only presented

evidence is included in Figure 6e, but this experiment is done in *Nicotiana*, and the magnitude of the effect between CCA1 and LHY is not even two fold.

(5) Data in Supplemental Figure 5c and 5d are not relevant, and in my opinion can be eliminated from the paper. The associated discussion in lines 280-285 are too speculative. If something, it only open more questions that requires additional analysis. It would be enough a brief discussion relating the timing of action of the evening complex (and of PRR repressors) compared to the new described SHB1/CCA1/LHY complex.

(6) Figure 2 and Supplementary Figure 2: Authors claim "In contrast, a *shb1-D cca1 lhy* triple mutant showed a hypocotyl phenotype and PIF4 expression similar to that of the *cca1 lhy* double mutant (Fig. 2d, e and Supplementary Fig. 2f). Thus, the promotion of hypocotyl and PIF4 expression by SHB1 under red light critically relies on CCA1 and LHY." While I agree that these data are consistent with the model that SHB1 requires CCA1/LHY, I think that formally there is an alternative explanation, that *shb1-D cca1 lhy* mutant do not show elevated levels of SHB1. How are the levels of SHB1 in the *shb1-D cca1 lhy* triple mutant compared to that of *shb1-D*?

(7) Supplemental Figure 2b-2c: Authors claim that no phenotypic or molecular effect is observed in *shb1 lhy cca1* triple mutant, compared to *cca1 lhy* double mutant. Because the original *shb1* mutant is in the *Col* genetic background, and the original *lhy cca1* mutant is in the *Ws* background, I wonder how the authors have controlled the contribution of the genetic background in establishing the phenotype of the hybrids compared to the parentals (i.e. the "Col Ws" line is actually much taller in red than parental *Col* and *Ws* lines).

(8) Figure 1d-e: Protein quantification is normalized over a ponceau S stained band. Because such a band could correspond to a protein which levels change during photomorphogenesis (presumably Rubisco), I don't think that this is a good normalization control. Either normalize to a better control (e.g. actin, as in supplemental figure 4), or consider this experiment as a "quantitative estimation" that then can be transferred to the supplemental material. Also the meaning of error bars in figure 1e. is not indicated.

(9) Line 182: "Therefore, the data suggest that PIF4 may also function in an independent pathway to promote hypocotyl elongation." Not clear what this is referred to.

(10) Supplemental Figure 2a legend: Growth conditions are not indicated.

Reviewers' comments:

Reviewer #1 (Remarks to the Author):

Line 48-56: The two back-to-back papers in Cell 1998 describing the original cloning of the first plant clock genes, CCA1 and LHY, deserve citation in this manuscript.

Revised from lines 53 to 56.

Line 449: "In general, CCA1 and LHY repress the evening element-containing genes at dawn," The original work discovered CCA1 as an activator of a dawn gene LHCB, this should be mentioned.

Revised from lines 447 to 448.

Reviewer #3 (Remarks to the Author):

(1) Authors acknowledged that PIF4 expression is regulated by the evening complex, yet the well documented direct repression of PIF4 expression by PRR components of the circadian clock is not mentioned. Appropriate references must be included.

Added as "Through CHIP quantitative PCR assays and genome-wide expression profiling, PRR5, PRR7 and PRR9 also bind to the upstream regions of *PIF4* and other key transcription factor genes, and repress their expression (Nakamichi et al., 2012)" from lines 278 to 281.

(2) The molecular lesion of the *shb1* mutant allele must be clarified. On the basis of new qRT-PCR data (Supplemental Figure 2e) authors conclude that "shb1 was a partial loss-of-function allele instead of a knockout allele.", as they observe that "In *shb1*, a T-DNA is inserted at the 8th intron of the *SHB1* gene (SALK_128406), and a truncated message was still produced (Supplementary Fig. 2d, e)."

However, no evidence for the presence of such a "truncated message" is provided. Authors determine by RT-PCR the presence of the 5' region of the transcript upstream of the T-DNA insertion, but this is no proof of a truncated transcript. Moreover, as the T-DNA insertion is located in an intron, there is the formal possibility that the intron is spliced correctly resulting in functional mature transcript. This point is particularly relevant as most of the functional evidence regarding *SHB1* relies either in a gain of function mutant (*shb1-D*) or in *SHB1* overexpression lines. The *shb1* mutant allele certainly shows consistent reduced levels of PIF4 transcript, but no significant phenotypic growth consequences are observed in the present study. Is there evidence of *shb1* mutant allele being a partial loss-of-function allele in other reported phenotypic responses?

Evidence for the presence of such a truncated *SHB1* message in *shb1* is provided in Supplementary Figure 2e.

The *shb1* mutant allele also showed a partial loss-of-function phenotype in endosperm proliferation and cellularization (Zhou et al., 2009; O'Neill et al., 2019). This statement is added from lines 186 to 188.

(3) Figures 1-3: The headlines of figure legends emphasize the role of SHB1 and CCA1 on different responses. However, genetic data are always performed with *cca1 lhy* double mutant, and therefore, the relative role of CCA1 and LHY is not assessed. Therefore, the headlines may not arbitrarily focus on CCA1, and must also include LHY (e.g. CCA1/LHY).

Revised as suggested.

(4) Moreover, in Discussion (line 433): “Our study suggests that the highly expressed circadian clock protein CCA1 in the morning recruits SHB1 to activate red light-induced PIF4 expression and desensitize light responses. LHY may play a less prominent role in this process.”

I don’t think there is strong evidence in the article that suggests that LHY has a minor role compared to CCA1. In order to make this claim, it would need a closer look. The only presented evidence is included in Figure 6e, but this experiment is done in *Nicotiana*, and the magnitude of the effect between CCA1 and LHY is not even two fold.

Revised as “Our study suggests that the highly expressed circadian clock proteins CCA1 and LHY in the morning recruits SHB1 to activate red light-induced PIF4 expression and desensitize light responses.”

(5) Data in Supplemental Figure 5c and 5d are not relevant, and in my opinion can be eliminated from the paper. The associated discussion in lines 280-285 are too speculative. If something, it only open more questions that requires additional analysis. It would be enough a brief discussion relating the timing of action of the evening complex (and of PRR repressors) compared to the new described SHB1/CCA1/LHY complex.

Figure 5c and 5d were deleted as suggested. Only a brief discussion remains from lines 278 to 283.

(6) Figure 2 and Supplementary Figure 2: Authors claim “In contrast, a *shb1-D cca1 lhy* triple mutant showed a hypocotyl phenotype and PIF4 expression similar to that of the *cca1 lhy* double mutant (Fig. 2d, e and Supplementary Fig. 2f). Thus, the promotion of hypocotyl and PIF4 expression by SHB1 under red light critically relies on CCA1 and LHY.”

While I agree that these data are consistent with the model that SHB1 requires CCA1/LHY, I think that formally there is an alternative explanation, that *shb1-D cca1 lhy* mutant do not show elevated levels of SHB1. How are the levels of SHB1 in the *shb1-D cca1 lhy* triple mutant compared to that of *shb1-D*?

We don’t have any polyclonal anti-SHB1 antibodies left available. We examined the expression of *SHB1* in *shb1-D* and *shb1-D cca1 lhy* triple mutant (Supplementary Fig. 2g). The levels of *SHB1* transcripts are comparable in *shb1-D* and *shb1-D cca1 lhy*. See lines 200 to 202.

(7) Supplemental Figure 2b-2c: Authors claim that no phenotypic or molecular effect is observed in *shb1 lhy cca1* triple mutant, compared to *cca1 lhy* double mutant. Because the original *shb1* mutant is in the Col genetic background, and the original *cca1 lhy* mutant is in the Ws background, I wonder how the authors have controlled the contribution of the genetic

background in establishing the phenotype of the hybrids compared to the parentals (i.e. the “Col Ws” line is actually much taller in red than parental Col and Ws lines).

The “Col Ws” line is indeed much taller in red than parental Col and Ws lines due to hybrid vigor. The cross of *shb1* in Col to *cca1 lhy* in Ws may also produce hybrid vigor. Several Col Ws lines were derived from the same cross when each *shb1*, *cca1* or *lhy* allele was genotyped and showed consistent phenotypes. One representative line was shown (added in Supplemental Figure 2 legend).

(8) Figure 1d-e: Protein quantification is normalized over a ponceau S stained band. Because such a band could correspond to a protein which levels change during photomorphogenesis (presumably Rusbico), I don't think that this is a good normalization control. Either normalize to a better control (e.g. actin, as in supplemental figure 4), or consider this experiment as a “quantitative estimation” that then can be transferred to the supplemental material. Also the meaning of error bars in figure 1e. is not indicated.

We normalized to actin blots with the same protein extracts. The error bars in Figure 1e are standard errors.

(9) Line 182: “Therefore, the data suggest that PIF4 may also function in an independent pathway to promote hypocotyl elongation.” Not clear what this is referred to.

Deleted.

(10) Supplemental Figure 2a legend: Growth conditions are not indicated.

Added in legend: Plants were grown in a growth room with a 16L/8D cycle at 22 °C under 30 $\mu\text{mol}/\text{m}^2/\text{s}$ fluorescent white light.

REVIEWERS' COMMENTS:

Reviewer #3 (Remarks to the Author):

Authors have addressed all my concerns.

Minor additional comments:

Line 292: "Surprisingly, the CCA1 ChIP peak did not coincide with the CCA1 mRNA peak (Supplementary Fig. 1b and Fig. 4d)."

The figure cited must be Supplementary Fig. 1c?

Fig. 4e: Ws background is indicated in the figure, whereas Col is indicated in the legend. This discrepancy must be resolved.

Supplemental Figure 5 legend heading "LUX or NOX and CCA1 or LHY have nearby binding sites in the PIF4 promoter." needs to be updated as no LUX nor NOX are currently analyzed.

REVIEWERS' COMMENTS:

Reviewer #3 (Remarks to the Author):

Minor additional comments:

1) Line 292: "Surprisingly, the CCA1 ChIP peak did not coincide with the CCA1 mRNA peak (Supplementary Fig. 1b and Fig. 4d)."

The figure cited must be Supplementary Fig. 1c?

Revised.

2) Fig. 4e: Ws background is indicated in the figure, whereas Col is indicated in the legend. This discrepancy must be resolved.

Revised.

3) Supplemental Figure 5 legend heading "LUX or NOX and CCA1 or LHY have nearby binding sites in the PIF4 promoter." needs to be updated as no LUX nor NOX are currently analyzed.

Revised as ELF4:GFP associates with the *PIF4* promoter.